# PREFGEN: MULTIMODAL PREFERENCE LEARNING FOR PREFERENCE-CONDITIONED IMAGE GENERATION

## ABSTRACT

Preference-conditioned image generation seeks to adapt generative models to individual users, producing outputs that reflect personal aesthetic choices beyond the given textual prompt. Despite recent progress, existing approaches either fail to capture nuanced user preferences or lack effective mechanisms to encode personalized visual signals. In this work, we propose a multimodal framework that leverages multimodal large language models (MLLMs) to extract rich user representations and inject them into diffusion-based image generation. We train the MLLM with a preference-oriented visual question answering task to capture fine-grained semantic cues. To isolate preference-relevant features, we introduce two complementary probing tasks: inter-user discrimination to distinguish between different users, and intra-user discrimination to separate liked from disliked content. To ensure compatibility with diffusion text encoders, we design a maximum mean discrepancy-based alignment loss that bridges the modality gap while preserving multimodal structure. The resulting embeddings are used to condition the generator, enabling faithful adherence to both prompts and user preferences. Extensive experiments demonstrate that our method substantially outperforms strong baselines in both image quality and preference alignment, highlighting the effectiveness of representation extraction and alignment for personalized generation.

## 1 INTRODUCTION

Generative models have rapidly advanced (Sohl-Dickstein et al., 2015; Saharia et al., 2022; Esser et al., 2024; Sauer et al., 2024), achieving remarkable progress in diversity, semantic fidelity, and visual realism, and have been widely applied in tasks such as text-to-image generation (Mo et al., 2024; Ba et al., 2025) and image editing (Hertz et al., 2023a; Han et al., 2023; Mo et al., 2025b; Lin et al., 2025). Alongside these advances, there has been growing interest in preference-conditioned image generation, as users often have different preferences even for images generated from the same prompt. To approximate their desired outcomes, users frequently refine prompts iteratively by adding modifiers and adjusting generation parameters. However, this manual process is labor-intensive and time-consuming, and it still falls short of capturing individual preferences that are complex, abstract, or difficult to express with text alone.

Psychological research suggests that aesthetic preferences are not arbitrary but are shaped by both low-level perceptual cues, such as color and contrast, and high-level semantic attributes, such as subject matter and composition (Iigaya et al., 2021). This implies that personal taste is partially explainable and can be inferred from observable visual properties. Such findings motivate a data-driven approach: if preferences can be compressed into structured representations, they can be extracted from a small set of reference images and used to condition generative models.

Popular generative models, such as unified vision-language models (Wu et al., 2024; Ma et al., 2024; Chen et al., 2025), have shown strong capabilities in both understanding and generation, making them suitable for tasks that require joint reasoning over visual and textual modalities. However, their ability to follow instructions for multi-image preference reasoning is limited, and their generation quality lags behind modern diffusion backbones for preference-conditioned image synthesis. Diffusion-based methods address this gap through two main strategies. Some directly perform style transfer from user-provided images (Ye et al., 2023; Wang et al., 2024; Hertz et al., 2023b), while others decouple the process into preference feature extraction followed by conditional generation. For example,

ViPer (Salehi et al., 2024) relies on user-provided descriptive text refined by large language models to inject preferences into the generation pipeline. More recent approaches leverage multimodal large language models (MLLMs) (Li et al., 2024; Liu et al., 2024) to aggregate multiple image references into robust embeddings for diffusion-based personalization (Song et al., 2024; Dang et al., 2025). Within this paradigm, IP-Adapter (Ye et al., 2023) offers a parameter-efficient mechanism for preference injection by exploiting CLIP's strong image–text alignment (Radford et al., 2021), outperforming alternative modules such as GLIGEN (Li et al., 2023) and ControlNet (Zhang et al., 2023) in both efficiency and fidelity. Despite these advances, aligning MLLM-derived embeddings with diffusion text encoders remains challenging. Recent studies attempt to bridge this gap with point-wise alignment objectives: MoMA (Song et al., 2024) uses mean squared error (MSE) loss to align MLLM embeddings with CLIP image embeddings, MIND-Edit (Wang et al., 2025) adopts cosine similarity, and CINEMA (Deng et al., 2025b) combines both to balance magnitude and directional alignment. While these objectives partially reduce the discrepancy, they often impose rigid constraints, limiting stability and hindering optimal transfer of preference information.

In light of these observations, several challenges remain. First, how can user preferences be effectively inferred from limited historical images. Second, how can these inferred preferences be encoded into representations capturing both semantic and stylistic information. Third, how can such representations be integrated into diffusion backbones in a way that ensures compatibility and stable conditioning.

To address these challenges, we propose a user-specific conditioning framework. First, we employ an MLLM pretrained on a preference-oriented visual question answering task to capture both semantic and stylistic cues from multiple user reference images. Second, we design two probing tasks to systematically identify preference-relevant features within the MLLM. Inter-user discrimination locates intermediate-layer features that differentiate users, forming a core identity embedding, while intra-user preference discrimination isolates high-level features that distinguish liked from disliked samples within a user, producing a semantic preference embedding. Third, to align MLLM-derived semantic preferences with the text embeddings expected by diffusion models, we introduce a maximum mean discrepancy (MMD)-based alignment loss (Gretton et al., 2012; Li et al., 2015). This loss maps semantic preference embeddings through a multilayer perceptron into the generator's text space, ensuring distributional compatibility while preserving multimodal structure. On the generative side, we adopt a lightweight conditioning injection using a decoupled cross-attention branch, which allows the model to remain faithful to text prompts while flexibly adjusting style and detail according to user embeddings. Our contributions are fourfold:

- We introduce a user-specific conditioning framework using MLLMs, trained on a preference-oriented VQA dataset to infer latent user preferences from few examples.
- We design two probing tasks to systematically analyze MLLM embeddings, disentangling cross-user identity signals from intra-user semantic preference signals and demonstrating their complementary roles in personalization.
- We develop an MMD-based distribution alignment loss that bridges the representational gap between semantic preference embeddings and the text space of diffusion backbones, improving compatibility and stability during generation.
- We construct PREFBENCH, a benchmark for evaluating preference personalization, and show empirically that our method achieves higher preference alignment and quality than baselines.

## 2 METHOD

We present PREFGEN, a novel framework that learns to generate preference-conditioned images from sparse user preference feedback, as shown in Fig. 1. Our framework seamlessly integrates multimodal preference modeling with conditional diffusion generation through three core innovations: (1) hierarchical preference embedding extraction via systematic MLLM analysis, (2) distribution-aligned preference conditioning, and (3) joint preference-semantic generation training.

### 2.1 PROBLEM FORMULATION

We formulate preference-conditioned image generation as a synthesis problem guided by both textual prompts and user-specific preferences. For a user $u$ with a sparse preference set $\mathcal{H}_u = \{(\mathbf{x}_i, \mathbf{y}_i)\}_{i=1}^n$, where $\mathbf{x}_i$ denotes an image and $\mathbf{y}_i \in \{0, 1\}$ indicates whether the image is disliked or liked, the

Figure 1: Overview of our framework. Step 1: Fine-tune an MLLM on preference-oriented VQA. Step 2: Perform layer analysis to extract identity embedding $\mathbf{e}_{\text{core}}$ and semantic preference embedding $\mathbf{e}_{\text{sem}}$. Step 3: Align $\mathbf{e}_{\text{sem}}$ with the text encoder space using an MMD loss, producing $\hat{\mathbf{e}}_{\text{sem}}$. Step 4: Inject $\hat{\mathbf{e}}_{\text{sem}}, \mathbf{e}_{\text{core}}, \mathbf{e}_{\text{img}}$ into the base model via cross-attention for preference-conditioned generation.

objective is to generate an image $\mathbf{x}^*$ for a given prompt $p$ that maximizes the probability of being preferred: $\mathbf{x}^* = \arg\max_{\mathbf{x}} P(\mathbf{y} = 1 \mid \mathbf{x}, p, \mathcal{H}_u)$. The generated image should simultaneously preserve semantic fidelity to the prompt and align with user-specific visual preferences, such as favored color palettes, styles, compositions, or object arrangements inferred from $\mathcal{H}_u$.

We hypothesize that user preferences, though diverse and subjective, can be compressed into a compact embedding that captures their essential patterns. Formally, we posit the existence of a mapping $f_\theta : \mathcal{H}_u \to \mathbb{R}^d$, with $d \ll |\mathcal{H}_u|$, producing a user embedding $\mathbf{e}_u = f_\theta(\mathcal{H}_u)$ that encapsulates the discriminative preference signals required to guide generation.

## 2.2 MULTIMODAL USER PREFERENCE MODELING

**Multimodal training.** The first stage aims to strengthen the MLLM's ability to infer user preferences from historical images and candidate content. To this end, we construct a preference-oriented VQA dataset $\mathcal{D}_{\text{pref}} = \{(\mathcal{H}_u, \mathbf{q}_j^{(u)}, \mathbf{a}_j^{(u)}, \mathbf{x}_j^{(u)}, \mathbf{y}_j^{(u)})\}$, where $\mathbf{x}_j^{(u)}$ denotes a candidate image, $\mathbf{q}_j^{(u)}$ is a preference-related query asking whether user $u$ would like $\mathbf{x}_j^{(u)}$, $\mathbf{y}_j^{(u)}$ denotes corresponding answers, and $\mathbf{a}_j^{(u)}$ represents the underlying attributes that drive the decision, such as artistic style, color palette, or composition.[1] The model parameters $\theta$ are optimized with cross-entropy loss. This training enables the model to build cross-modal associations between user histories, visual attributes, and preference judgments. We denote the resulting model as PREFDISC, which functions as a preference discriminator.

**Layer-wise probing for preference signals.** An important step in our framework is to identify which MLLM layers best capture user preferences. We freeze the pretrained parameters $\theta$ and attach lightweight probes (linear classifiers) to embeddings from different layers. These probes are evaluated through two complementary tasks: (i) *preference discrimination* and (ii) *multi-user identification*.

For the preference discrimination task, we construct *paired* examples for each user $u$. Each pair consists of a preferred image and a non-preferred image that shares the same prompt. We extract embeddings $\mathbf{e}_j^{(u)} = f_\theta(\mathcal{H}_u, \mathbf{q}_j^{(u)}, \mathbf{x}_j^{(u)})$ and train a logistic regression $h_\phi : \mathbb{R}^d \to [0, 1]$ with the standard cross-entropy loss:

$$\mathcal{L}_{\text{binary}}(\phi) = -\frac{1}{|\mathcal{D}_{\text{sub}}|} \sum_{(\mathbf{e}_j^{(u)}, \mathbf{y}_j) \in \mathcal{D}_{\text{sub}}} [\mathbf{y}_j \log \sigma(h_\phi(\mathbf{e}_j^{(u)})) + (1 - \mathbf{y}_j) \log(1 - \sigma(h_\phi(\mathbf{e}_j^{(u)})))]. \quad (1)$$

For the multi-user identification task, we assess whether embeddings preserve user-specific preference signatures by training a $U$-way classifier $g_\psi : \mathbb{R}^d \to \Delta^{U-1}$ with cross-entropy loss:

$$\mathcal{L}_{\text{multi}}(\psi) = -\frac{1}{|\mathcal{D}_{\text{sub}}|} \sum_{(\mathbf{e}_j^{(u)}, u) \in \mathcal{D}_{\text{sub}}} \log P_\psi(u \mid \mathbf{e}_j^{(u)}). \quad (2)$$

---

[1]In early experiments, we attempted to predict both $\mathbf{y}_j^{(u)}$ and $\mathbf{a}_j^{(u)}$ jointly, but this led to unstable outputs. Thus, attributes are not modeled in this stage.

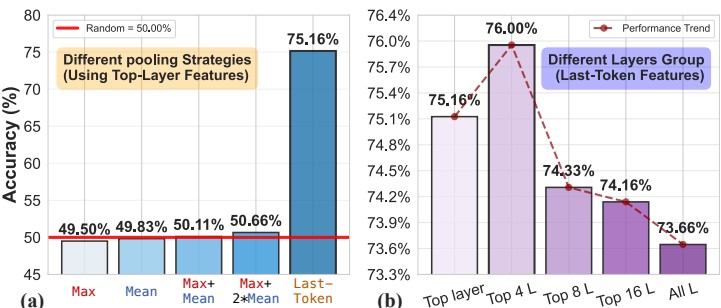

Table 1: Study comparing layer selections on the 300-user identification task. The results show that using embeddings $\mathbf{e}_{core}$ from the middle layers achieves the best performance.

| Layer | Strategy | $\text{Acc}_{\times 10^3}$ |
|---|---|---|
| Random | – | 3.30% |
| Top layer | Last Token | 36.70% |
| Top 4 layers | Last Token | 40.00% |
| Layers 8-24 | Last Token | 43.30% |
| Layers 8-20 | Last Token | **63.30%** |
| Top 16 layers | Last Token | 40.00% |
| All layers | Last Token | 40.00% |

Figure 2: Study comparing different pooling strategies and layer selections for the preference discrimination task. The results show that using embeddings $\mathbf{e}_{sem}$ from the top four layers with the last-token strategy achieves the best performance in like–dislike discrimination.

To conduct this analysis, we construct a balanced subset $\mathcal{D}_{\text{sub}} \subset \mathcal{D}_{\text{pref}}$. Then we extract embeddings from different regions of a 32-layer MLLM: the final layer, shallow ranges covering the top 4 or 8 layers, deeper ranges covering the top 16 layers, and the full architecture. For multi-layer settings, embeddings from selected layers are summed. To handle variable-length token sequences, we systematically compare five *pooling strategies*: `Last-Token`, `Max` pooling, `Mean` pooling, and hybrid strategies `Max+Mean`, `Max+2×Mean`.

**Layer probing insights.** Our analysis reveals a hierarchical structure of preference representations in MLLMs, offering practical guidance for modeling user-specific signals. (1) *Effect of pooling.* As shown in Fig.2(a), the `Last-Token` strategy consistently outperforms others. In contrast, `Max`, `Mean`, or their combinations tend to dilute informative cues. The last token instead serves as a multimodal summary that concentrates the most discriminative information. (2) *Where preference emerges.* In Fig.2(b), the preference discrimination task achieves its best accuracy when using embeddings from the top four layers. These layers encode high-level semantics and surface-level stylistic cues (e.g., color tone, rendering style, lighting patterns) that are essential for distinguishing preferences between images generated from the same prompt. We denote these embeddings as $\mathbf{e}_{sem} \in \mathbb{R}^d$. (3) *Where identity persists.* In Tab. 1, the more challenging user identification task peaks (about $20\times$ above random baseline) when embeddings are taken from middle-to-upper layers. These intermediate representations capture stable identity markers and core aesthetic tendencies that generalize across diverse prompts and contents. We denote them as $\mathbf{e}_{core} \in \mathbb{R}^d$.

> **Main takeaway:** Top layers capture semantic and stylistic cues for preference discrimination ($\mathbf{e}_{sem}$), while middle-to-upper layers encode stable identity traits ($\mathbf{e}_{core}$). Pooling with the `Last-Token` is consistently the most effective.

## 2.3 Distribution Alignment via Maximum Mean Discrepancy Loss

Building on these findings, we incorporate both $\mathbf{e}_{core}$ and $\mathbf{e}_{sem}$ as conditional signals for generative models. A central challenge arises from the representational gap between MLLM-derived embeddings and the pre-trained generative backbones, which are typically conditioned on text-only embeddings (e.g., CLIP). While both $\mathbf{e}_{core}$ and $\mathbf{e}_{sem}$ deviate from the text embedding distribution, we focus alignment efforts on $\mathbf{e}_{sem}$. The rationale is that $\mathbf{e}_{sem}$ encodes high-level semantic and stylistic abstractions that are directly comparable to textual embeddings, as discussed in Sec. 2.2.

To bridge this gap, $\mathbf{e}_{sem}$ is first transformed by a 6-layer MLP into $\hat{\mathbf{e}}_{sem}$. We then minimize the Maximum Mean Discrepancy (MMD) (Gretton et al., 2012; Li et al., 2015) between $\hat{\mathbf{e}}_{sem}$ and the paired CLIP text embeddings $\mathbf{e}_{text}$. The latter is obtained from the last token representation of the CLIP text encoder applied to the underlying preference attributes $\mathbf{a}^2$. Empirically, this last-token embedding provides a compact yet informative summary of the entire input sentence, making it a robust target for semantic alignment. Formally, given sets of semantic embeddings $\{\hat{\mathbf{e}}_{sem}\}$ and their paired text embeddings $\{\mathbf{e}_{text}\}$, the MMD loss is defined as:

$$\mathcal{L}_{\text{MMD}} = \mathbb{E}_{\hat{\mathbf{e}}_{sem}, \hat{\mathbf{e}}'_{sem}}[k(\hat{\mathbf{e}}_{sem}, \hat{\mathbf{e}}'_{sem})] + \mathbb{E}_{\mathbf{e}_{text}, \mathbf{e}'_{text}}[k(\mathbf{e}_{text}, \mathbf{e}'_{text})] - 2\,\mathbb{E}_{\hat{\mathbf{e}}_{sem}, \mathbf{e}_{text}}[k(\hat{\mathbf{e}}_{sem}, \mathbf{e}_{text})],$$
$$(3)$$

---

[2] The CLIP text embeddings $\mathbf{e}_{\text{text}}$ are derived from the attribute descriptions $\mathbf{a}_j^{(u)}$ introduced in Sec. 2.2.

where $(\cdot)'$ denotes an independently sampled embedding from the same distribution, and $k(\cdot, \cdot)$ is a universal kernel. In practice, we adopt the Gaussian Kernel (Schölkopf & Smola, 2002).

*Why distributional alignment?* Point-wise losses such as MSE or cosine similarity force embeddings to match specific anchors, which our ablations (Sec. 3.4) show can over-constrain the model and cause collapse under limited data. In contrast, $\mathcal{L}_{\text{MMD}}$ aligns the overall embedding distribution, preserving preference diversity while ensuring compatibility with the text space.

## 2.4 Conditional Generation with Unified User Representation

After aligning the semantic embedding, we further strengthen user representations by integrating fine-grained visual cues. To this end, we introduce $\mathbf{e}_{img}$, an embedding derived from a user-provided *liked* image, obtained through the CLIP image encoder. This embedding grounds the representation in low-level visual patterns and complements the semantic abstraction of $\mathbf{e}_{sem}$ and the stable identity traits of $\mathbf{e}_{core}$. Empirically, the inclusion of $\mathbf{e}_{img}$ not only accelerates training convergence but also produces outputs that are more coherent and faithful to user intent.

**Fusion strategies.** The final user representation is constructed by concatenating the three components: $\mathbf{e}_u = [\hat{\mathbf{e}}_{sem}; \mathbf{e}_{core}; \mathbf{e}_{img}]$, where $\hat{\mathbf{e}}_{sem}$ is the distribution-aligned semantic embedding, $\mathbf{e}_{core}$ encodes stable user identity traits, and $\mathbf{e}_{img}$ provides fine-grained visual anchoring. We explored several fusion strategies (Fig. 12): `Attn` (cross-attention), `Res-Attn` (attention with residual connection), and `Concat` (direct concatenation). Although attention-based methods are more flexible, they add complexity and instability. `Concat` is both simpler and more effective in our ablations, suggesting that a straightforward combination is sufficient in our case.

**Conditioning mechanism.** The concatenated representation $\mathbf{e}_u$ is injected into the generative backbone using the IP-Adapter framework (Ye et al., 2023). Specifically, in each text cross-attention layer, we add a parallel user cross-attention branch: the query $\mathbf{Q}$ attends to both text-derived $(\mathbf{K}, \mathbf{V})$ and user-derived $(\mathbf{K}', \mathbf{V}')$, and the outputs are combined additively: $\text{Attention}(\mathbf{Q}, \mathbf{K}, \mathbf{V}) + \text{Attention}(\mathbf{Q}, \mathbf{K}', \mathbf{V}')$.

**Training objective.** The overall training objective extends the standard score-matching loss:

$$\mathcal{L}_{total} = \mathbb{E}_{t,\mathbf{z}_0,\boldsymbol{\epsilon}} \left[ \| \boldsymbol{\epsilon} - \boldsymbol{\epsilon}_\theta(\mathbf{z}_t, t, p, \mathbf{e}_u) \|^2 \right], \tag{4}$$

where $p$ is the text prompt, $\mathbf{z}_t$ the noisy latent at timestep $t$, and $\boldsymbol{\epsilon}$ the injected noise. While we instantiate this loss with SDXL, the formulation is agnostic to the backbone. In principle, $\mathbf{e}_u$ can be incorporated into other generative models such as DiT-based flow matching models (e.g., FLUX (Black Forest Labs, 2024b), SD3 (Esser et al., 2024)), which we leave as future work.

# 3 Experiments

## 3.1 Data Collection

Our study builds on two complementary datasets that provide scalability and authenticity. The first dataset is constructed using preference-simulating agents, following the design principle of multi-dimensional visual attributes (Salehi et al., 2024). We employ Claude-3.5-Sonnet (Anthropic, 2024) as the backbone agent and assign each agent a personalized aesthetic profile. These profiles are defined across multiple visual dimensions and consist of both liked and disliked attributes. To ensure both diversity and internal consistency, each agent is assigned about fifty attributes on average. During data generation, we sample a random subset of the entire attributes and append them to a human-written prompt (Wang et al., 2023; Lin et al., 2014), guiding the generative process with agent-specific preferences. Images are then generated using the FLUX.1-dev model (Black Forest Labs, 2024a). After generation, agents annotate the results by marking which images align with or deviate from their preferences, yielding at least ten like–dislike pairs per agent. This pipeline results in a large-scale dataset of roughly one million images (990,998) from over fifty thousand simulated users (50,153). To prevent information leakage, we partition the dataset by user identity: 80% of the users for training, 10% for validation, and the remainder for testing. For evaluation, we further sample 136 users from the test set, each providing queries constructed from ten annotated images, forming the PREFBENCH benchmark. To complement this synthetic dataset with real user signals, we curate a second dataset from Pick-a-Pic (Kirstain et al., 2023), which provides genuine

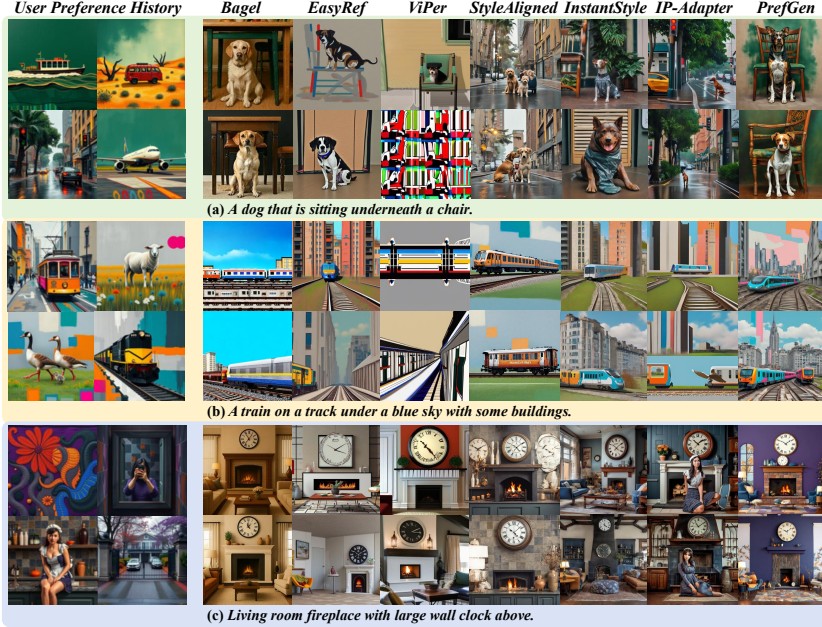

| *User Preference History* | *Bagel* | *EasyRef* | *ViPer* | *StyleAligned* | *InstantStyle* | *IP-Adapter* | *PrefGen* |

(a) *A dog that is sitting underneath a chair.*

(b) *A train on a track under a blue sky with some buildings.*

(c) *Living room fireplace with large wall clock above.*

Figure 3: Qualitative comparison with different methods. Each row shows the user's preference and outputs from different approaches. PREFGEN consistently captures both stylistic and semantic aspects of user preference, while others often fail to balance preference alignment and prompt fidelity.

human preference annotations. We begin by collecting image–text pairs associated with unique user IDs and grouping them into user-specific clusters. We then refine the data through automated filtering, removing image–text pairs with low alignment scores (Radford et al., 2021) as well as images inconsistent with broadly shared human preferences (Kirstain et al., 2023). After this process, the dataset contains 3,838 images from 347 distinct users in the test set. Additional details and examples are provided in App. B. To better evaluate PREFGEN on real human preferences, we further conducted generation experiments on the Pick-a-Pic dataset, which contains 317,682 real images from 2,301 distinct users. We sampled 100 preference sets from 38 real users, each consisting of five like–dislike pairs and one additional user-annotated image for evaluation.

## 3.2 EXPERIMENTAL SETUP

**Evaluation metrics.** We evaluate our framework along two dimensions: (i) overall image quality and (ii) alignment with user preferences. For image quality, we adopt two widely used metrics. Fréchet Inception Distance (FID) (Heusel et al., 2017) and CMMD score (Jayasumana et al., 2024) which serves as a more robust alternative. Lower values on both indicate higher fidelity and realism. To assess preference alignment, we report four metrics. CLIP Image Score (Radford et al., 2021) quantifies semantic consistency by computing the similarity between generated images and user-preferred references in the CLIP embedding space. The CSD metric (Somepalli et al., 2024) evaluates whether the generated images preserve the artistic styles and aesthetic attributes reflected in user-preferred images. We further report PREFDISC, a classifier designed to distinguish user-preferred images from non-preferred ones, which provides a reliable estimate of preference discrimination. Its effectiveness as a proxy is validated in App. D.4. Finally, we conduct human evaluations to complement automatic metrics and directly assess perceptual alignment with user preferences.

**Comparison to other methods.** We compare against both single-reference and multi-reference personalization methods. For single-reference methods, we include IP-Adapter (Ye et al., 2023), which injects a single reference image into diffusion models through decoupled cross-attention, serving as a standard approach for image-conditioned generation. InstantStyle (Wang et al., 2024) is a strong baseline for capturing user-preferred styles. It extends IP-Adapter by selectively modulating cross-attention layers to achieve more precise style transfer. StyleAligned (Hertz et al., 2023b) performs consistent style generation through a straightforward inversion operation. Bagel (Deng et al., 2025a) is a unified instruction-driven editing framework that adapts to diverse user requirements, representing a flexible reference point for instruction-based personalization. For multi-reference

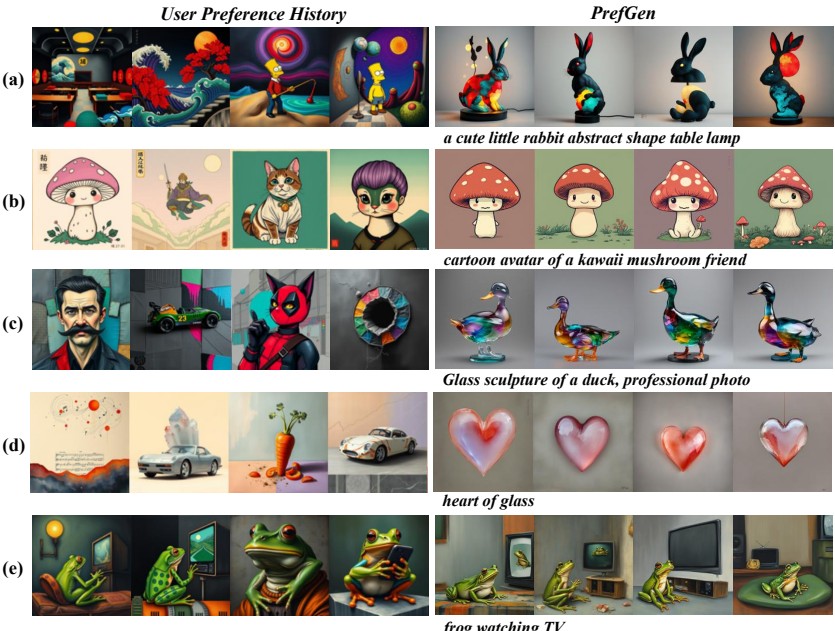

Figure 4: Application examples of PREFGEN. (a) Product design: PREFGEN integrates the color composition extracted from a user's preference history into the design of a rabbit-shaped lamp, aligning the generated output with user-specific aesthetic demands. (b) Character design: Given visual attribute specifications, PREFGEN generates characters with distinct background colors while preserving the desired attributes.

methods, we compare with ViPer (Salehi et al., 2024), which encodes user likes and dislikes into textual embeddings and guides generation accordingly, thus representing preference modeling via textual abstraction. We also compare with EasyRef (Zong et al., 2024), which leverages multiple reference images to capture consistent visual attributes through multimodal representation learning.

Table 2: Quantitative comparison of different methods on the PREFBENCH dataset.

| Method | FID ($\downarrow$) | CMMD ($\downarrow$) | CLIP Img ($\uparrow$) | CSD ($\uparrow$) | PREFDISC ($\uparrow$) |
|---|---|---|---|---|---|
| IP-Adapter | 172.06 | **0.25** | 67.88 | 50.22 | 81.65 |
| Instant Style | 162.20 | 0.37 | 73.40 | 54.69 | 79.09 |
| StyleAligned | 167.54 | 0.31 | 67.89 | 46.88 | 79.71 |
| ViPer | 210.89 | 0.75 | 68.04 | 39.94 | 54.57 |
| Bagel | 148.92 | 0.33 | 72.39 | 50.84 | 66.82 |
| EasyRef | 157.48 | 0.26 | 74.23 | 54.62 | 69.19 |
| PREFGEN (Ours) | **143.79** | **0.25** | **76.03** | **59.22** | **81.86** |

## 3.3 EVALUATION AND ANALYSIS

**Qualitative analysis.** Fig. 3 presents a qualitative comparison with several methods. In (a), the user history indicates a preference for painterly textures and muted, green-toned palettes. Our method successfully preserves these stylistic cues, producing images with brushstroke-like rendering and harmonious background hues. In contrast, IP-Adapter tends to drift toward the semantics of the reference image rather than the intended preference, while InstantStyle and StyleAligned inject only marginal stylistic signals. Bagel and EasyRef reproduce coarse semantic attributes but overlook the nuanced artistic style. ViPer further fails to maintain semantic coherence, leading to distorted structures. In (b), the history shows a bias toward abstract, block-like compositions and vibrant, segmented color fields. Our method integrates these artistic structures while maintaining the correct semantic form of the train. PREFGEN can infer a user's preferred artistic style from a small set of reference images. In practice, each user can construct a painting profile by specifying a few preference-indicating images. Once established, this profile guides subsequent generations to remain faithful to the chosen style. In Fig. 4, this enables diverse applications such as (a) product design, (b) character design, (c) creative image generation, (d) visual ideation, and (e) character cloning.

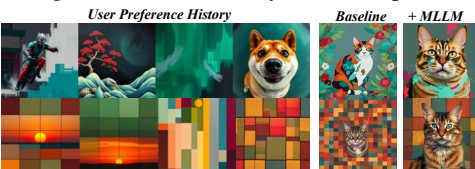

Figure 5: Evaluation by human experts.

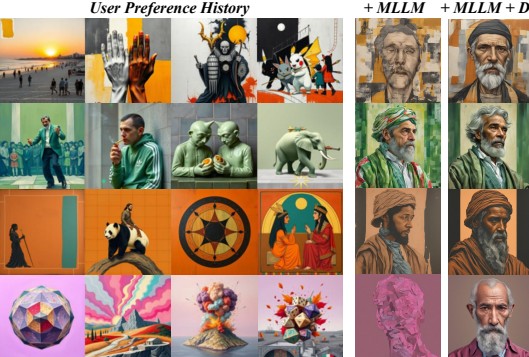

Figure 6: Ablation study on MLLM embeddings with the prompt "A cat".

Figure 7: Ablation study on the effect of distribution alignment with the input prompt "A man".

**Quantitative comparison.** As shown in Tab. 2, across all metrics, our method PREFGEN consistently achieves superior results on PREFBENCH. In terms of image quality, PREFGEN obtains the lowest FID and CMMD score, indicating that our method produces more realistic samples that remain faithful to the distribution of user-preferred images. For preference alignment, our approach achieves the highest CLIP Img score, demonstrating stronger semantic consistency with user-preferred images than other methods. On CSD, PREFGEN outperforms all methods, which suggests that our method allows the model to retain fine-grained aesthetic cues. Finally, PREFDISC confirms that our generation's best reflects user-specific judgments, surpassing even IP-Adapter's strong score by achieving 81.86. In Tab. 3, PREFGEN maintains strong preference alignment on real user data, with CLIP-Img, CSD, and PREFDISC scores remaining well above all baselines. This shows that PREFGEN generalizes beyond synthetic agents and performs reliably on real human preferences.

Table 3: Quantitative comparison of different methods on the processed Pick-a-Pic dataset containing real user preference histories.

| Method | FID (↓) | CMMD (↓) | CLIP Img (↑) | CSD (↑) | PREFDISC (↑) |
|---|---|---|---|---|---|
| IP-Adapter | 206.96 | 0.27 | 0.68 | 44.81 | 68.94 |
| Instant Style | 189.40 | 0.26 | 0.76 | 54.46 | 69.40 |
| StyleAligned | 200.31 | 0.25 | 0.68 | 41.30 | 68.39 |
| ViPer | 225.63 | 0.50 | 0.70 | 46.10 | 53.61 |
| Bagel | 190.11 | **0.24** | 0.71 | 49.35 | 56.22 |
| EasyRef | 199.34 | 0.25 | 0.74 | 51.09 | 65.80 |
| PREFGEN (Ours) | **189.34** | 0.25 | **0.77** | **57.13** | **76.78** |

**User study.** To further evaluate preference alignment from a human perspective, we conduct a user study with 15 experts who have prior experience in visual design and aesthetic evaluation. Each expert is presented with pairs of images generated by different models under the same textual prompt and identical user history. Their task is to select the image that better reflects the preferences implied by the reference set. The evaluation focuses on two aspects: first, whether the generated image semantically matches the textual description, and second, whether its color palette, artistic style, and overall visual presentation are consistent with the aesthetic signals exhibited in the user's liked images. As shown in Fig. 5, our approach achieves clear superiority across all comparisons, with win rates exceeding 63% against every method. These results demonstrate that PREFGEN not only maintains semantic fidelity to the prompt but also captures subtle stylistic cues from user histories.

**t-SNE visualization of MLLM feature representations.** To examine whether the learned embeddings $\mathbf{e}_{core}$ and $\mathbf{e}_{sem}$ capture user-specific preference structures, we project them into a two-dimensional space using t-SNE. We extract embeddings from 170 preference histories in the test set, covering 17 users. Each point in Fig. 9 and Fig. 10 corresponds to a user's preference history embedding derived from multiple labeled images, and points are color-coded by user identity. The visualization shows that embeddings of preference histories from the same user form coherent clusters, while embeddings from different users remain well separated. This demonstrates that the model encodes consistent signals of individual preference rather than collapsing to generic representations. Moreover, we present a comparison of the $\mathbf{e}_{sem}$ and $\mathbf{e}_{core}$ in App. D.6.

Table 4: Ablation study on embeddings and alignment loss.

| $\mathbf{e}_{img}$ | $\mathbf{e}_{sem}$ | $\mathbf{e}_{core}$ | Alignment Loss | FID ($\downarrow$) | CMMD ($\downarrow$) | CLIP Img ($\uparrow$) | CSD ($\uparrow$) | PrefDisc ($\uparrow$) |
|---|---|---|---|---|---|---|---|---|
| ✓ | | | - | 151.65 | 0.64 | 72.76 | 45.03 | 59.77 |
| ✓ | ✓ | ✓ | - | 152.84 | 0.32 | 75.30 | 54.44 | 74.68 |
| | ✓ | | $\mathcal{L}_{MMD}$ | 163.35 | 0.41 | 64.25 | 31.16 | 55.98 |
| | ✓ | ✓ | $\mathcal{L}_{MMD}$ | 159.72 | 0.41 | 69.21 | 37.47 | 56.83 |
| ✓ | ✓ | ✓ | $\mathcal{L}_{MSE} + \mathcal{L}_{cos}$ | 150.34 | 0.50 | 70.94 | 43.36 | 57.97 |
| ✓ | ✓ | ✓ | $\mathcal{L}_{MMD}$ | **143.79** | **0.25** | **76.03** | **59.22** | **81.86** |

Table 5: Swapping-based analysis of the embeddings.

| Method | CLIP Text ($\uparrow$) | CSD ($\uparrow$) |
|---|---|---|
| PREFGEN (Ours) | 25.83 | 59.22 |
| Swap $\mathbf{e}_{sem}$ | 25.56 | 58.29 |
| Swap $\mathbf{e}_{img}$ | 25.76 | 53.27 |
| Swap $\mathbf{e}_{core}$ | 25.63 | 58.99 |

## 3.4 ABLATION STUDY

**Effect of MLLM embeddings.** We examine the contribution of MLLM embeddings by comparing the baseline model with its variant augmented by preference-aware representations. As illustrated in Fig. 6, the inclusion of MLLM features enables the generator to more faithfully capture user history. Quantitative results in Tab. 4 confirm this observation: although the FID remains close to the baseline, the preference-sensitive metrics show clear improvements, with notable gains in CLIP Img, CSD, and PREFDISC scores, indicating stronger alignment with user-specific aesthetic preferences.

**Disentanglement Analysis via Latent Swapping.** In Tab. 5, we conduct controlled swapping and ablation experiments under fixed prompts to examine whether the hierarchical decomposition captures functionally distinct factors. Replacing $\mathbf{e}sem$ results in the largest degradation in the CLIP-Text score, decreasing from 25.83 to 25.56, indicating its primary role in semantic alignment. Replacing $\mathbf{e}img$ leads to the most severe drop in CSD, decreasing from 59.22 to 53.27, confirming its role in fine-grained stylistic control. In contrast, replacing $\mathbf{e}_{core}$ produces consistent but distributed degradation across both metrics, suggesting that it encodes stable, user-level preference traits that are orthogonal to purely semantic or purely stylistic factors. In Fig. 8, we visualize a controlled swapping experiment where $\mathbf{e}_{sem}$, $\mathbf{e}_{img}$ and $\mathbf{e}_{core}$ are exchanged across users, revealing their distinct roles in semantic alignment, stylistic consistency, and stable user-level aesthetic modeling.

**Effect of distribution alignment.** We further investigate the impact of distribution alignment. In Fig. 7, incorporating DA produces generations that are more coherent and stylistically consistent with the reference set. In Tab. 4, the combined model (+MLLM+DA) achieves the best overall performance across all metrics. In particular, FID and CMMD scores improve significantly over both the baseline and the +MLLM variant, while CLIP Img and CSD continue to increase, demonstrating enhanced semantic fidelity and style preservation.

**Ablation on distribution alignment loss.** We compare the effect of point-wise losses such as Mean Squared Error (MSE) and Cosine Similarity against $\mathcal{L}_{MMD}$. The combined loss is formulated as $\mathcal{L} = \alpha \cdot \mathcal{L}_{point} + \beta \cdot \mathcal{L}_{MMD}$, where $\mathcal{L}_{point}$ represents the combination of MSE and cosine similarity loss, and $\alpha, \beta$ are coefficients. In Fig. 11, we present the generation results at different training steps under the setting of limited data. When $\beta = 0$, training often collapses, leading to severe degradation in generation quality after a few steps. This occurs because point-wise alignment forces individual embeddings to match specific targets, which over-constrains the model and causes it to lose preference-relevant information extracted from the user history. The quantitative results in Tab. 4 further show that $\mathcal{L}_{MMD}$ consistently outperforms point-wise objectives across all preference-oriented metrics, while also yielding the best FID and CMMD scores. By contrast, using only $\mathcal{L}_{MMD}$ produces much more stable results, as MMD aligns the overall distribution of embeddings without forcing one-to-one matches, thereby preserving the global preference structure. When both objectives are combined, the two losses tend to conflict: point-wise objectives pull embeddings toward specific anchor points, while MMD encourages them to follow the global text embedding distribution.

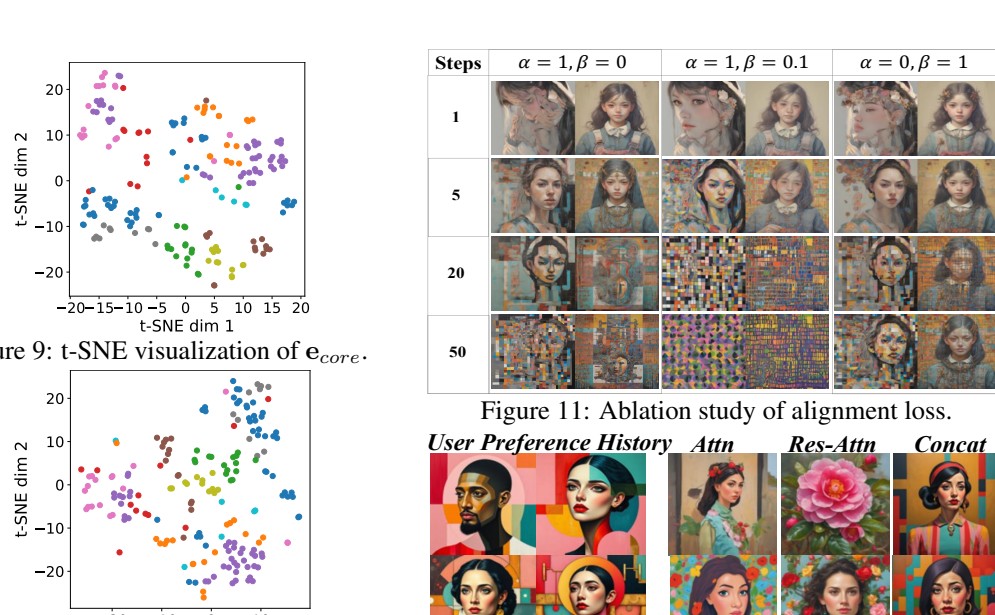

Figure 8: We conduct a controlled embedding swapping experiment across two users, where the preference histories of user (a) and user (b) are used to extract three embeddings: $\mathbf{e}_{sem}$, $\mathbf{e}_{img}$, and $\mathbf{e}_{core}$. In (c), we present the generated images under different swapping settings. For each generation, one component extracted from (a) is replaced by the corresponding component from (b), while keeping the other components fixed.

Figure 9: t-SNE visualization of $\mathbf{e}_{core}$.

Figure 10: t-SNE visualization of $\mathbf{e}_{sem}$.

Figure 11: Ablation study of alignment loss.

Figure 12: Ablation study of fusion strategies.

**Fusion strategies for user preference integration.** Fig. 12 compares three fusion strategies for incorporating user preference embeddings. `Attn` employs cross-attention, where the CLIP embedding $\mathbf{e}_{img}$ serves as the query and $\mathbf{e}_{core}$ and $\hat{\mathbf{e}}_{sem}$ serve as keys and values. `Res-Attn` builds on `Attn` by adding a residual connection with a zero-initialized linear projection and a subsequent layer normalization. `Concat` directly concatenates the embeddings, forming a single fused representation that is injected into the UNet. Compared with the attention-based strategies, concatenation proves to be more straightforward and effective. This suggests that for our scale preference modeling, a simple concatenation can sufficiently capture user-aligned features.

## 4 CONCLUSION

In this work, we introduced PREFGEN, a framework for preference-conditioned image generation that leverages multimodal large language models to extract user-specific representations and incorporates a distribution alignment mechanism to bridge the gap between multimodal embeddings and diffusion backbones. By disentangling stable identity traits from context-dependent semantic preferences and aligning the latter to the generator's text space, our method produces images that faithfully reflect both prompt semantics and individual aesthetic tastes. Extensive experiments, including human expert evaluations, demonstrate that PREFGEN consistently surpasses existing personalization methods in image quality and preference alignment.

## Reproducibility Statement

The Use of LLM is detailed in App. E. All models and training setups used with complete hyperparameter configurations in our work are described in App. C. Dataset statistics and preprocessing steps are documented in Sec. 3.1 and App. B to facilitate replication. To comply with institutional policies and maintain double-blind review, we will release implementation resources only after the review period. Upon publication, we will release the full codebase, including data preprocessing scripts and experiment pipelines, under a license to enable full reproducibility.

## Ethics Statement

This work adheres to the ICLR Code of Ethics. All datasets used in this study are publicly available or synthetically generated and do not contain personally identifiable information. Our data collection process respects the terms of use of the source platforms, and all preference annotations were simulated or automatically derived without involving human subjects in a way that would require IRB approval. The proposed method is for research purposes to improve personalization in generative models and does not intend to promote harmful or biased content. We will release our code and data under a research-friendly license to ensure transparency, reproducibility, and responsible use.

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

In this supplementary material, we provide additional details and analyses that complement the main paper. We first review relevant literature, including prior work on unified vision-language models and concept-conditioned generation, as well as studies on personalized preference extraction, multimodal conditioning, and personalization strategies for diffusion models such as fine-tuning, token-based methods, and adapters. We then describe the datasets used in our experiments, including the processing steps applied to Pick-a-Pic and the construction of the agent dataset. Next, we outline further experimental details, with a particular focus on the impact of the number of reference images. We also present additional experiments, covering evaluations of preference discrimination, the influence of training data quality, computational efficiency, and resource usage.

## A  RELATED WORK

Research on Preference-Conditioned and multimodal generation spans several directions. One line of work develops unified vision-language models that close the gap between understanding and generation. Another thread focuses on extracting and injecting user-specific signals into pretrained generative backbones. A third direction explores lightweight or training-free personalization techniques that trade computational cost for deployment flexibility. We summarize these areas below and position our contributions accordingly.

**Unified VLMs and concept-conditioned generation.**   Recent efforts have sought to build unified vision-language models (Wu et al., 2024; Ma et al., 2024; Chen et al., 2025) and semantic encoders that can support both recognition and generation. These models emphasize learning high-quality semantic features that can be reused across tasks, rather than training separate image priors or autoencoders for each domain. UniCTokens (An et al., 2025) extends this approach toward personalization by learning unified concept tokens that bridge personalized understanding and generation, improving both concept recognition and synthesis. Our work shares the motivation of reusing semantic signals from multimodal encoders, but differs by explicitly separating stable user identity traits from context-dependent semantic preference representations and aligning the latter to the text embedding space of generative backbones.

**Personalized preference extraction and multimodal conditioning.**   Another growing research direction focuses on extracting user preferences and transforming them into effective conditioning signals for generative models. ViPer (Salehi et al., 2024) presents a structured pipeline for preference-centric personalization by collecting user comments on a small set of images and converting them into attribute-level constraints for generation, underscoring the importance of structured preference capture. Personalized Multimodal Generation explores the integration of behavioral traces with multimodal embeddings for large language model driven personalization, showing that combining symbolic and embedding-based preferences enhances downstream generation quality. (Mo et al., 2025a) introduces a contrastive learning framework based on MLLMs to effectively capture fine-grained user preferences. By modeling relative preference relationships among samples, this approach distinguishes between user "likes" and "dislikes." UniCTokens (An et al., 2025) and EasyRef (Zong et al., 2024) also contribute to this line of research by proposing mechanisms to integrate multi-image references and user-provided concepts into unified VLMs and diffusion models. EasyRef (Zong et al., 2024) leverages multimodal large language models to aggregate multi-image references into robust and generalizable embeddings for diffusion conditioning, demonstrating how consistent signals can be distilled across multiple reference images for plug-and-play personalization. Our approach similarly extracts multi-image preference cues using MLLMs, but we explicitly decompose the outputs into two representations: a core identity embedding that captures stable and generalizable traits, and a semantic preference embedding that reflects context-dependent decisions. We preserve the identity embedding as a separate conditioning pathway and further align the semantic preference embedding with the generator's text space to reduce representational mismatch.

**Fine-tuning, token-based, and adapter personalization for diffusion models.**   Personalization in text-to-image diffusion models has also been studied through parameter-efficient fine-tuning, token-level injection, and adapter-based conditioning. Textual Inversion (Gal et al., 2023) and DreamBooth (Ruiz et al., 2023) represent early and influential methods that associate a subject or concept with a newly learned token or fine-tuned model weights, thereby enabling faithful synthesis of the subject in diverse contexts. While effective, these methods require per-concept optimization and

careful strategies to avoid overfitting. IP-Adapter (Ye et al., 2023) introduces a modular adapter design that injects image prompts into pretrained diffusion models without retraining the backbone, enabling flexible image-conditioned personalization. More recent work such as Subject-Diffusion (Ma et al., 2023) and DreamSteerer (Yu et al., 2024) extends these ideas to improve subject-driven generation and editability, supporting identity preservation and flexible editing with minimal additional training. Compared with these approaches, our method combines the stability benefits of image-anchored conditioning with an explicit semantic alignment stage that maps MLLM-derived semantic preference embeddings into the text-conditioning manifold used by generative backbones, thereby enabling more robust and user-aligned generation. Unlike subject-driven personalization methods, which focus on binding a generator to a specific instance, our work tackles user-preference personalization, where the goal is to capture stable and context-dependent aesthetic preferences from a few reference images and use them to guide generation.

## B  DATASET

Our training and evaluation rely on two complementary sources of data. The first is a curated subset of the Pick-a-Pic dataset (Kirstain et al., 2023), which provides real user annotations. The second is a synthetic dataset generated by agents designed to simulate diverse user preferences.

**Pick-a-Pic Dataset Processing.**  We leverage the Pick-a-Pic dataset, which contains image–text pairs annotated by real users. To ensure data quality and minimize noise introduced by irrelevant prompts, we apply a multi-step filtering pipeline. First, we compute CLIP text embeddings and cluster them using DBSCAN (Ester et al., 1996), which naturally identifies semantic groups without requiring the number of clusters to be specified. Prompts identified as outliers are removed. Next, we prune samples that fail to meet general quality standards based on both PickScore and CLIPScore, using empirically chosen thresholds. Finally, approximately 10% of remaining samples that still exhibit unclear or inconsistent preferences are manually discarded. This procedure yields a cleaned subset that better reflects user preference signals while reducing annotation noise. Some examples are shown in Figure 13.

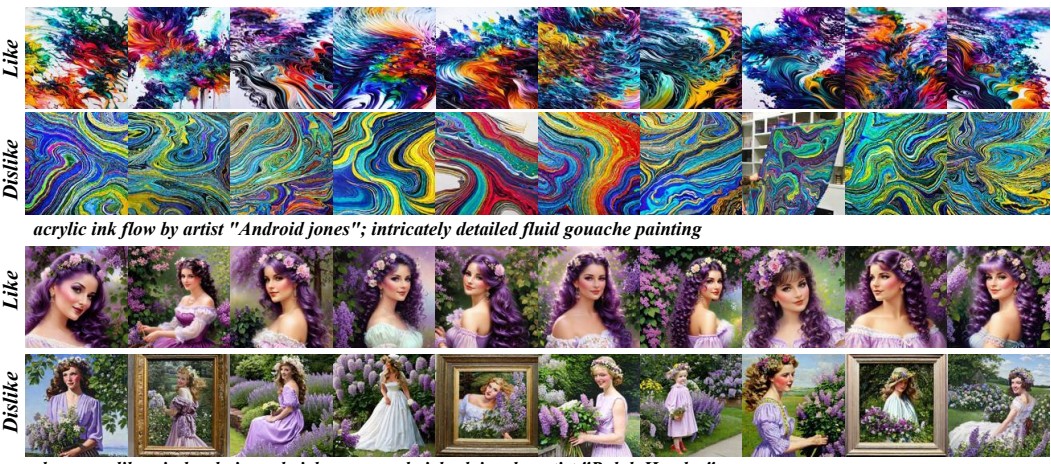

*acrylic ink flow by artist "Android jones"; intricately detailed fluid gouache painting*

*she wears lilacs in her hair, and picks roses and picks daisys by artist "Ralph Horsley"*

Figure 13: Some examples of the processed Pick-a-Pic dataset.

**Agent Dataset Construction.**  Following (Salehi et al., 2024) and (Mo et al., 2025a), we construct artificial users with distinct and consistent aesthetic preferences. To this end, we adopt the Claude-3.5-Sonnet agent (Anthropic, 2024) and employ the Flux.1-dev (Black Forest Labs, 2024a) generator to synthesize images. We begin by defining a comprehensive taxonomy of aesthetic attributes that spans several key dimensions shown in Table 6, which inspired by (Salehi et al., 2024), including artistic style, color palette, compositional strategy, skill level, level of detail, dominant hues, and medium. Each agent is assigned a personalized profile consisting of preferred and disliked attributes sampled from this taxonomy. This configuration yields controllable, fine-grained, and highly heterogeneous

preferences, thereby approximating the diversity observed in real-world users. The input prompt to Claude-3.5-Sonnet is as follows:

```
You are a user with a distinct aesthetic preference profile.
Your profile is defined across multiple visual dimensions such as
artistic style, color palette, composition strategy, level of detail,
dominant hues, and medium.
From this taxonomy, a subset of attributes has been assigned to you.
These attributes are divided into two categories:
- Liked attributes: <attr>, <attr>, ..., <attr>
- Disliked attributes: <attr>, <attr>, ..., <attr>
Treat these preferences as stable and consistent, reflecting your unique
aesthetic identity.

You will be given a text prompt for image generation and a set of
generated images.
Your task is to evaluate each image according to the following procedure:

1. Examine the image carefully and compare it against your assigned
liked and disliked attributes.
2. Decide whether the image aligns with your liked attributes or
conflicts with your disliked attributes.
3. Assign a binary label to the image:
   - If the image matches your liked attributes, label it as "liked."
   - If the image matches your disliked attributes or deviates from your
taste, label it as "disliked."
4. Ensure consistency across all evaluations by relying strictly on the
given profile, without adding any personal judgment beyond the assigned
attributes.
5. For each image, output only "liked" or "disliked," without additional
commentary.

Here is the base text prompt for image generation: <prompt>
Here is the generated image: <image>
Here is the next image: <image>
```

Some examples of the dataset are shown in Figure 14. Moreover, $\mathcal{D}_{\text{sub}} \subset \mathcal{D}_{\text{pref}}$ (in Sec. 2.2) containing 300 users, each associated with 11 labeled images.

**The Processed ImageNet and ImageNet-sketch Dataset.** To evaluate whether PrefGen generalizes to user preferences outside the training distribution, we construct an OOD preference discrimination task using real photographs and sketches from ImageNet (Deng et al., 2009) and ImageNet-Sketch (Wang et al., 2019), which are never seen during training. We use PickScore to annotate preference pairs and test whether our learned preference discriminator remains consistent with a high-quality external preference oracle on real data. Some examples are shown in Figure 15.

## C EXPERIMENTAL DETAILS

Our experiments are designed to accommodate varying numbers of preference history images, following the observations in (Salehi et al., 2024) regarding the relationship between reference count and preference discrimination. For each user, we randomly sample 6–14 annotated images as conditions, with all images resized to $512 \times 512$. For PREFDISC, we follow (Salehi et al., 2024) and use IDEFICS2-8B (Laurençon et al., 2024) as the multimodal backbone. Training runs for 3,000 steps on the training split with eight A100 (80GB) GPUs, an effective batch size of 16, a learning rate of $1 \times 10^{-5}$, and weight decay $1 \times 10^{-2}$. The distribution alignment module is a lightweight MLP (see Appendix C.1), trained for 400k steps on one A100 GPU with AdamW, learning rate $1 \times 10^{-4}$, and batch size 64. For PREFGEN, we pre-extract embeddings from the MLLM to reduce computational overhead during training. The weights of the MLLM are kept frozen, and only the adapter layers in the diffusion model, including the newly introduced decoupled cross-attention branch and its corresponding projectors, are trained. Stable Diffusion XL (Podell et al., 2024) is used as the base generator, initialized with IP-Adapter weights. Training uses AdamW with batch

| Dimension | Representative Attributes |
|---|---|
| Artistic Medium | Painting (Oil, Fresco, Acrylic, Gouache), Sculpture (Stone, Metal, Ceramic, Glass), Textile Arts (Weaving, Quilting, Embroidery), Digital Forms (3D Modeling, AI Art, Virtual Reality), Printmaking (Etching, Woodcut, Silkscreen), Photography, Ceramics, Drawing |
| Color Palettes | Vibrant (Radiant Red, Vivid Purple), Pastel Hues (Lilac, Blush Pink, Powder Blue), Earthy Colors (Olive Green, Warm Brown, Clay Gray), Dark Colors (Charcoal Black, Deep Indigo, Burgundy), Oceanic Tones (Aquamarine, Navy, Turquoise), Neon Shades (Electric Lime, Atomic Green), Autumnal (Pumpkin Spice, Cranberry Red) |
| Skill Expression | Dynamic, Intuitive, Meticulous, Gestural, Bold, Spontaneous, Polished/Raw, Inventive, Lyrical, Controlled, Experimental, Graceful/Powerful, Free-flowing/Structured |
| Compositional Strategy | Rhythmic patterns, Centralized forms vs. decentralized layouts, Foreground and background contrast, Shallow depth vs. extended depth, Open areas vs. enclosed zones, Figure–ground interplay, Balanced or unbalanced arrangements, Installation format vs. pictorial surface, Fragmented structures vs. compressed ones, Static versus dynamic qualities, Positive and negative spatial tension, Contrasting divisions, Grid-structured organization, Exaggerated vs. flattened perception |
| Detail Treatment | Intricate, Simplified, Vivid, Subtle, Fine, Elaborate, Rough, Ethereal, Ambiguous, Textured, Realistic, Expressive, Minimalistic, Muted |
| Hues | Red, Scarlet, Magenta, Teal, Slate, Indigo, Gold, Copper, Yellow, Emerald, Azure, Turquoise, Burgundy, Cerulean, Crimson |
| Art Styles | Baroque Art, Graffiti, Impressionism, Ukiyo-e, Surrealism, Minimalism, Romanticism, Byzantine Art, Digital Art, Cubism, Mannerism, Conceptual Art, Renaissance Art, Futurism, Street Art, Symbolism, Art Nouveau, Queer Art, Contemporary Abstraction |

Table 6: A taxonomy of aesthetic attributes spanning style, medium, palette, composition, skill, detail, and hue.

size 16, learning rate $1 \times 10^{-4}$, weight decay $1 \times 10^{-2}$, and 300k steps. Supervision is provided by user-liked images. During inference, images are generated using 30 denoising steps with a guidance scale of 0.6, following the hyperparameter settings in (Ye et al., 2023). Results are averaged over five seeds $(1, 2, 3, 4, 5)$.

For training the proposed PREFDISC model, we adopt a quantized low-rank adaptation (QLoRA) strategy. Specifically, the LoRA configuration is set with rank $r = 8$, scaling factor $\alpha = 8$, and dropout rate $0.1$. We apply LoRA modules to all projection layers, including down-projection, gate-projection, up-projection, and the standard attention projections $(k, q, v, o)$, across the text encoder, modality projection, and perceiver resampler. The LoRA weights are initialized using a Gaussian distribution to ensure stable training. Our framework is implemented on top of PyTorch, and the

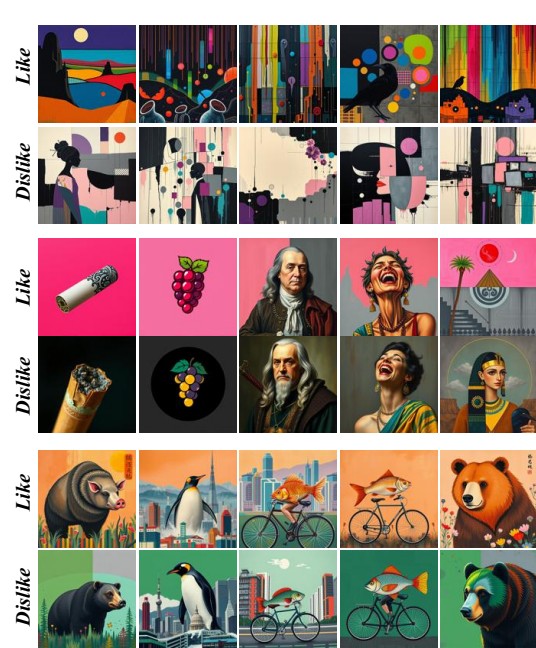

*Color Field Painting, De Stijl, Pixel Art, Lyrical Abstraction, Electric Lime, Laser Blue, Vivid Purple, Charcoal Black, Earthy Amber, Grid-based, Installation, Flattened, Figure-ground relationship, Golden Ratio, Rule of Thirds, Centralized, Deep space, Unpredictable, Impulsive, Restrained, Subtle, Heavy-handed, Bold, Rigorous, Graceful, Ethereal, Intricate, Rough, Ambiguous, Tactile, Unfocused, Abstracted, Minimalistic, Gold, Scarlet, Slate Gray, Indigo, Azure, Crimson, Turquoise, Etching, Woodcut, Encaustic, Embroidery, Macrame, Kinetic, Engraving*

*Situationist Art, Pop Art, Romanticism, Metaphysical Painting, Naive Art, Art of Ancient Egypt, Mesoamerican Art, Pink, Electric Lime, Radiant Red, Autumn Leaf, Alloy Silver, Olive Brown, Charcoal Gray, Concrete Gray, Background, Symmetry, Deep space, Balance, Contrasting Sections, Rule of Thirds, Open space, Static, Efficient, Eloquent, Fluid, Deliberate, Naive, Meticulous, Restrained, Bold, Selective, Unfocused, Focused, Rough, Intricate, Muted, Rose, Red, Scarlet, Teal, Found Object, Ink, Coiling, Woodcut, Chalk, Etching*

*Realism, Postmodern Art, Chinese Art, Naive Art, Stuckism, Rococo Art, De Stijl, Color Field Painting, Peach, Earthy Amber, Desert Rose, Deep Indigo, Cranberry Red, Antique Lavender, Blush Pink, Iron Black, Dynamic, Rule of Thirds, Flattened, Grid-based, Rhythmic, Deep space, Foreground, Raw, Dynamic, Expressive, Heavy-handed, Innovative, Impulsive, Subtle, Unpredictable, Sharp, Intricate, Rough, Soft, Simplified, Muted, Tactile, Realistic, Copper, Yellow, Turquoise, Azure, Slate, Orange, Red, Film, Earthenware, Virtual Reality, Polaroid, Silkscreen, Knitting, Watercolor, Found Object*

Figure 14: Some examples of the agent dataset.

Maximum Mean Discrepancy (MMD) loss is incorporated following the open-source implementation provided by Lee (2020).

For comparison, we follow the original settings of baseline methods. ViPer relies on explicit preference signals from users, formulated as feedback statements such as "I like this image." Bagle extracts user preferences from the provided reference images and generates outputs conditioned on these preferences, combined with the given prompt in the form: "Extract the user's preferences based on the reference images and generate an image that conforms to the user's preferences. The content is: <prompt>." EasyRef is evaluated with its default prompting strategy without additional modification.

## C.1 IMPLEMENTATION DETAILS OF PROJECTION AND ADAPTER MODULES

**Projection Model for $e_{core}$.** The projection model is a lightweight projection network that maps MLLM embeddings into additional context token for the generator's cross-attention layers. The module consists of a single linear projection followed by reshaping and LayerNorm. Specifically, MLLM embeddings of dimension $4096$ are projected into $k \times d_{attn}$ with $k = 1$ and $d_{attn} = 2048$. The resulting tensor is normalized before being used as extra context tokens.

**MLP Adapter for $e_{sem}$.** The MLP Adapter is a six-layer feedforward network designed to compress high-dimensional embeddings into a compact latent representation. The input dimension of $4096$ is first projected through a linear layer of size $4096$, followed by a LayerNorm, a GELU activation, and a dropout layer with a rate of $0.1$. After that, the network contains four hidden layers, each of size $2048$, with the same LayerNorm–GELU–dropout structure. Finally, the output is mapped to a 2048-dimensional representation and reshaped to $[B, 1, 2048]$ for integration into the generator.

## C.2 IMPACT OF THE NUMBER OF REFERENCE IMAGES

Prior works (Salehi et al., 2024; Mo et al., 2025a) have shown that incorporating as few as three liked and disliked images into the context already yields around 60% accuracy in preference prediction, and that performance continues to improve as the number of reference images increases. Building on this observation, we adopt a more flexible strategy during training. Instead of fixing the number

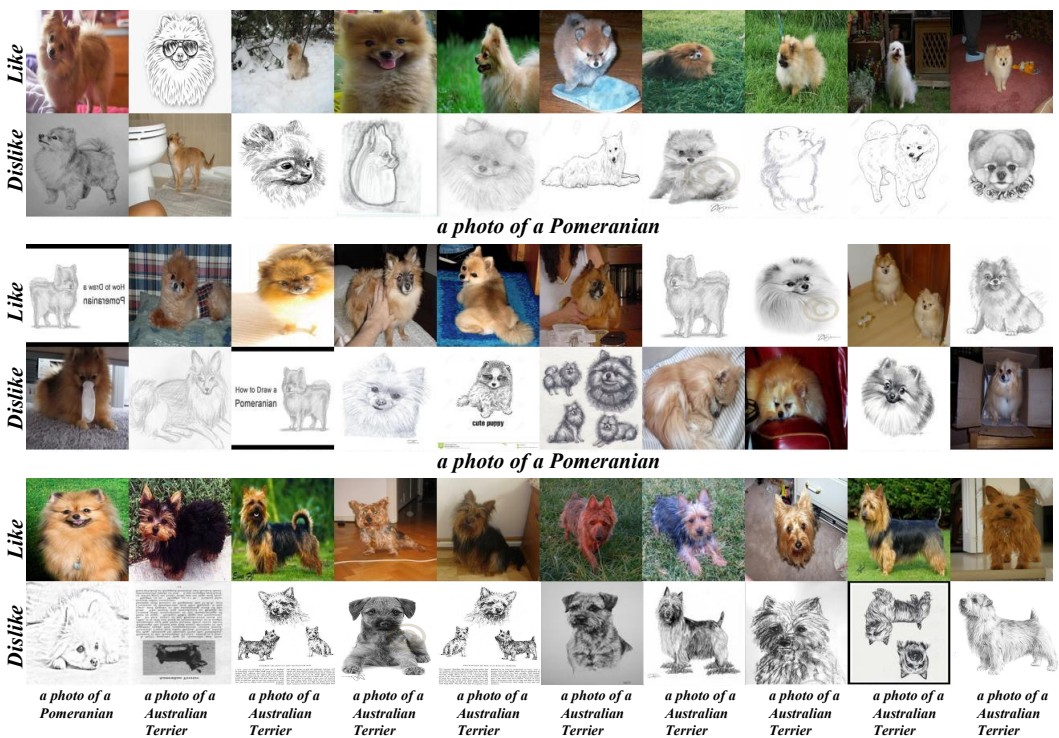

Figure 15: Some examples from the processed ImageNet (Deng et al., 2009) and ImageNet-Sketch (Wang et al., 2019) datasets for OOD preference evaluation. For each prompt, images are automatically grouped into Like and Dislike sets using PickScore. The Like set contains higher-scoring images, while the Dislike set contains lower-scoring samples.

of reference images, we randomly vary the input size to encourage robustness across different user scenarios. Specifically, for each user, we sample between 6 and 14 annotated images as conditions. This design choice allows the model to generalize better across varying history lengths and avoids overfitting to a fixed reference size.

In Figure 16, we compare ViPer and our PREFDISC across different sequence lengths. Our method remains stable, while ViPer proxy metric degrades when the sequence is either too short (insufficient preference information) or overly long (noisy or conflicting signals).

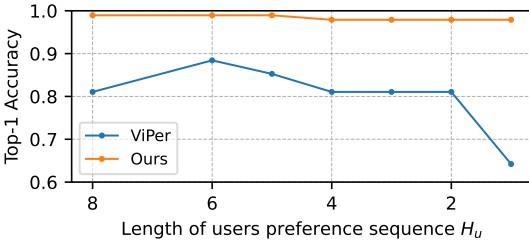

Figure 16: Effect of the length of the user preference sequence $H_u$ on PREFBENCH. We compare ViPer and our method across different sequence lengths. Our method remains stable while ViPer degrades when the sequence becomes overly long.

### C.3 COMPARISON TO OTHER METHODS.

Table 7 summarizes the number of reference images required and whether each method uses only positive examples or both positive and negative user preferences.

Table 7: Comparison of input requirements across preference-conditioned generation methods.

| Method | # Reference Images | Uses Positive Examples | Uses Negative Examples |
|---|---|---|---|
| IP-Adapter | Single | ✓ | ✗ |
| InstantStyle | Single | ✓ | ✗ |
| StyleAligned | Single | ✓ | ✗ |
| Bagel | Single | ✓ | ✗ |
| EasyRef | Multiple | ✓ | ✗ |
| ViPer | Multiple | ✓ | ✓ |
| PREFGEN (Ours) | Multiple | ✓ | ✓ |

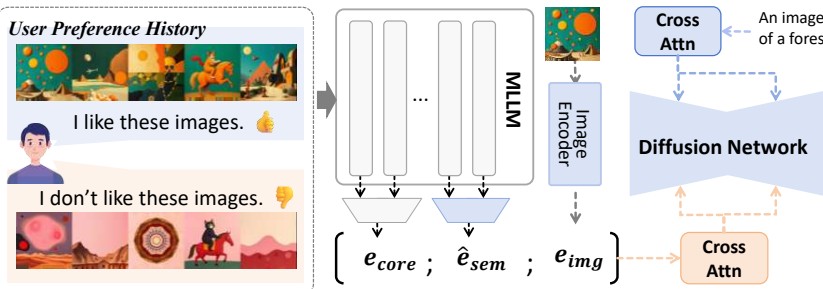

Figure 17: Overview of the inference-stage pipeline of PREFGEN.

### C.4 OVERVIEW OF INFERENCE STAGE.

As illustrated in Figure 17, at inference time, a user's preference history consisting of liked and disliked images is processed by a frozen MLLM. The intermediate representations are then mapped by two separate adaptation heads: a linear projection layer to obtain the core identity embedding $\mathbf{e}_{core}$, and a 6-layer MLP to obtain the semantic preference embedding $\hat{\mathbf{e}}_{sem}$. Implementation details of these projection adapters are provided in Appendix C.1. In parallel, a CLIP image encoder extracts a visual embedding $\mathbf{e}_{img}$ from a representative liked image. These embeddings are concatenated to form a unified user representation and injected into the frozen diffusion backbone via a decoupled cross-attention branch, together with the text embedding of the input prompt.

## D MORE EXPERIMENTS

### D.1 IMPACT OF TRAINING DATA QUALITY

To investigate the role of training data quality, we compare preference discrimination accuracy under different data configurations. ViPer constructs its proxy metric by combining agent-annotated data with the Pick-a-Pic dataset, which is not publicly available. In contrast, our agent data provides carefully curated annotations with substantially higher quality.

The results in Table 8 clearly demonstrate the impact of training data quality. PREFDISC trained solely on our agent data achieves the highest accuracy on PREFBENCH and also outperforms all other configurations on the Pick-a-Pic test set. These findings underscore the necessity of high-quality preference data for building effective discrimination models.

### D.2 OOD GENERALIZATION TO NEW USERS AND REAL-WORLD IMAGE TYPES.

We randomly sample 8 image pairs as user conditions and build 120 positive/negative pairs across ImageNet and ImageNet-Sketch. The evaluation reports discrimination accuracy, AUC, and statistical significance metrics. In Table 9, the results show significant alignment between PrefDisc's preference direction and PickScore's preference judgments, even when the input domain shifts from model-generated images to natural photographs and sketches. Because PrefDisc builds on inherently generalizable MLLM representations, leverages diffusion models that already encode broad visual

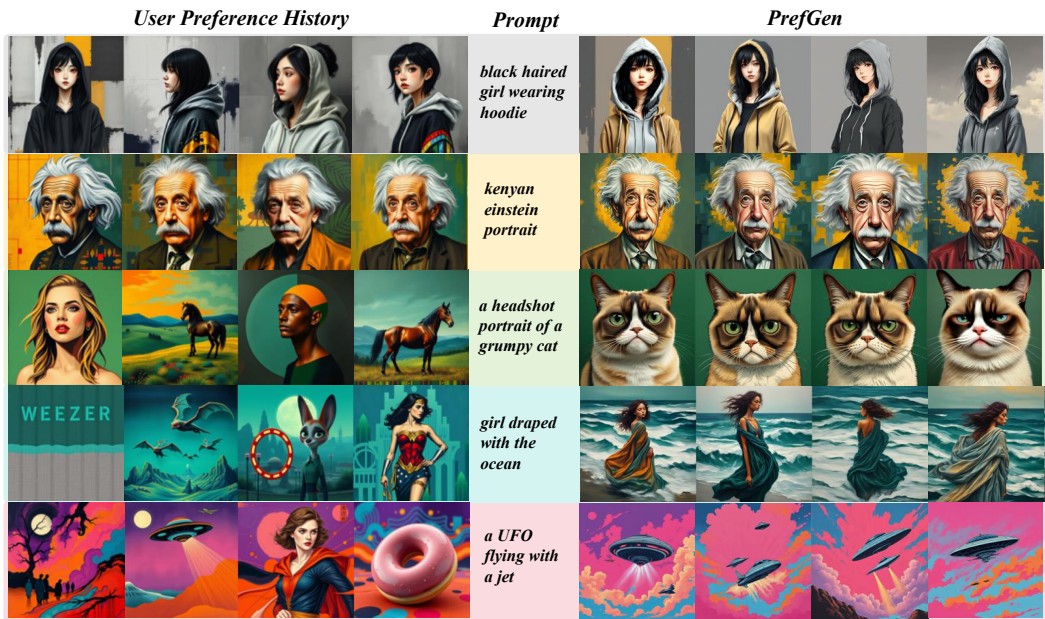

Figure 18: The images generated by PREFGEN. Each example shows user history preference on the left, the text prompt in the middle, and results from PREFGEN on the right. Our approach adapts to user-specific aesthetic signals, generating outputs that more faithfully reflect the preference history.

Table 8: Comparison of different training data on preference discrimination accuracy (%).

| Training Data | Model | Pick-a-Pic | PREFBENCH |
|---|---|---|---|
| Pick-a-Pic + ViPer's agent data | ViPer Proxy Metric | 52.74% | 81.62% |
| Pick-a-Pic | PrefDisc | 51.89% | 53.68% |
| Our agent data + Pick-a-Pic | PrefDisc | 52.11% | 85.29% |
| Our agent data | PrefDisc | **53.13%** | **98.67%** |

diversity, and adopts a training objective (distribution-level alignment) that preserves rather than distorts these foundational capabilities, the model naturally generalizes to user preferences beyond the training distribution.

Table 9: Quantitative results of OOD preference prediction.

| Metric | Value |
|---|---|
| Accuracy | 0.7583 (91 / 120) |
| AUC | 0.7552 |
| Wilcoxon p-value | <1e-6 (stat = 6038.00) |
| Paired t-test p-value | <1e-6 (t = 7.53) |
| Mean diff (pos–neg) | 0.3048 ± 0.4418 |
| Median diff | 0.2810 |

### D.3 MORE QUALITATIVE ANALYSIS RESULTS

In Figure 18, we present the results of five users for five prompts that explicitly define the art style, medium, or color palette. Even though the results are still personalized and there is a clear preference pattern in each row, they consistently adhere to the input prompt's meaning.

## D.4 PREFERENCE DISCRIMINATION EVALUATION

To further assess the capability of PREFDISC in preference estimation, we evaluate it on two benchmark datasets and report top-1 accuracy as the primary metric. The evaluation protocol involves both automatic agents and human annotators to ensure a fair comparison.

We begin with human evaluation, where ten experts are asked to infer user preferences. For every test case, annotators are presented with several reference images explicitly divided into liked and disliked examples, followed by two candidate images. Their task is to identify which candidate aligns more closely with the inferred preferences.

We evaluate automatic agents under the same conditions. For each test instance, the Claude agent is provided with the same reference images and candidate pair. The agent receives a structured prompt that specifies which images are liked and which are disliked, and is instructed to infer visual preference patterns before selecting the preferred candidate. The input prompt is:

```
You are given a set of reference images that represent a users
preferences.

Images in [<image>,...,<image>] are liked.
Images in [<image>,...,<image>] are disliked.

Your task is to:

1. Analyze the reference images to infer the users visual preference
patterns.
2. Compare the two candidate images: <image> and <image>.
3. Select the single candidate (index 0 or 1) that best matches the
inferred preferences.

Output format:
Return only the index (0 or 1) of the chosen candidate, followed by a
one-sentence explanation of the decision.
Do not output anything else.
```

Unlike ViPer Proxy Model, which is limited by a fixed number of reference examples, PREFDISC is designed to flexibly handle varying numbers of reference images. This adaptability makes it more practical for real-world personalization, where the amount of available preference data often varies across users. Furthermore, PREFDISC is trained on a broader and higher-quality dataset generated by Flux.1-dev (Black Forest Labs, 2024a), providing stronger coverage of visual preference patterns compared to the smaller, agent-limited datasets used by ViPer.

Table 10 presents the comparative results. On the processed Pick-a-Pic dataset, PREFDISC achieves performance on par with human experts and consistently surpasses both ViPer and Claude. On PREFBENCH, PREFDISC reaches 98.67%, effectively matching human-level discrimination while clearly outperforming all other automated baselines. These results highlight that PREFDISC provides a reliable estimate of user preference discrimination and can serve as a robust proxy for measuring preference alignment.

Table 10: Comparison of different methods on preference discrimination accuracy (%).

| Model | Pick-a-Pic | PREFBENCH |
|---|---|---|
| ViPer Proxy Metric | 52.74% | 81.62% |
| Claude-3.5-Sonnet | 47.97% | 90.44% |
| Human | 57.61% | 98.67% |
| PrefDisc | 53.13% | 98.67% |

## D.5 COMPUTATIONAL EFFICIENCY AND RESOURCE USAGE

We compare the computational requirements of our framework with existing methods along three axes: dataset scale, GPU usage, and training time. EasyRef is trained on over 130M high-quality

images together with 4M text–image pairs, which requires nearly five days of computation on 64 H800 GPUs. Bagel operates on an even larger corpus of 2,665M examples, although the paper does not disclose the exact GPU configuration or training duration. Both methods therefore, rely on massive datasets and substantial GPU resources. ViPer is trained on the Pick-a-Pic dataset and adopts a setup by constructing preference data from 5,000 agents, each annotated with 50 positive and negative attributes and corresponding generated images. Its proxy metric is trained in less than five hours on a single A100-80GB GPU, but the paper does not report the GPU usage or training time for its generative model.

Our method offers a balanced trade-off between efficiency and performance. PREFDISC is trained on approximately 1M preference-labeled examples and converges within one day using 8 A100-80GB GPUs, while PREFGEN requires about five days on the same hardware. This resource footprint is far smaller than that of EasyRef and Bagel, yet the resulting models deliver competitive or superior performance. Moreover, compared to ViPer, our approach delivers stronger performance in preference-conditioned image generation.

Table 11 reports the inference time to generate one image. PREFGEN requires only a single MLLM forward pass and achieves competitive latency (2.30 s), which is close to IP-Adapter (1.89 s) and InstantStyle (1.85 s). In contrast, StyleAligned is significantly slower due to the inversion steps, and ViPer incurs additional cost from multiple autoregressive text-generation passes. Table 12 summarizes GPU memory usage. A standard SDXL pipeline uses 6.57 GB, and IP-Adapter with SDXL requires 10.75 GB. Our method has a steady-state footprint of 26.50 GB, which is substantially lower than ViPer (42.24 GB) and comparable to Bagel (27.30 GB).

Table 11: Inference latency (wall-clock time in seconds) per image under same hardware settings.

| IP-Adapter | Instant Style | StyleAligned | Bagel | ViPer | EasyRef | PREFGEN |
|---|---|---|---|---|---|---|
| 1.89s | 1.85s | 27.45s | 18.71s | 5.66s | 3.50s | 2.30s |

Table 12: GPU memory (VRAM) usage of different methods during inference.

| Method | Base VRAM (Allocated) |
|---|---|
| SDXL (base) | 6.57 GB |
| IP-Adapter | 10.75 GB |
| PREFGEN | 26.50 GB |
| ViPer (w/ Refiner) | 42.24 GB |
| EasyRef | 11.68 GB |
| Bagel | 27.30 GB |

### D.6 COMPARING $\mathbf{e}_{sem}$ AND $\mathbf{e}_{core}$ REPRESENTATIONS

To better understand the functional roles of different embedding layers, we compare $\mathbf{e}_{core}$, extracted from middle-to-upper layers, with $\mathbf{e}_{sem}$, derived from the topmost layers. The former is expected to retain enduring user-specific traits, while the latter refines these traits into more explicit semantic judgments. To assess their clustering properties, we apply K-means (MacQueen, 1967) over embeddings from 17 categories (users) and evaluate the resulting clusters against ground-truth labels using several standard metrics.

As shown in Table 13, $\mathbf{e}_{core}$ consistently outperforms $\mathbf{e}_{sem}$ across NMI, AMI (Nguyen et al., 2010), V-measure, Homogeneity, and Completeness (Rosenberg & Hirschberg, 2007). These results indicate that $\mathbf{e}_{core}$ embeddings better capture global category structures, yielding clusters that are both more homogeneous within each class and more separated across different classes. This analysis suggests that $\mathbf{e}_{core}$ serves as a more faithful representation of underlying categorical information, making it a stronger candidate for downstream applications that require robust preference modeling.

Table 13: Clustering performance comparison between $\mathbf{e}_{sem}$ and $\mathbf{e}_{core}$. Higher values indicate better clustering quality.

| Metric | $\mathbf{e}_{sem}$ | $\mathbf{e}_{core}$ |
|---|---|---|
| ARI ($\uparrow$) | 0.3641 | 0.497 |
| NMI ($\uparrow$) | 0.6497 | 0.7024 |
| AMI ($\uparrow$) | 0.5332 | 0.6017 |
| Homogeneity ($\uparrow$) | 0.6538 | 0.7082 |
| Completeness ($\uparrow$) | 0.6456 | 0.6968 |
| V-measure ($\uparrow$) | 0.6497 | 0.7024 |

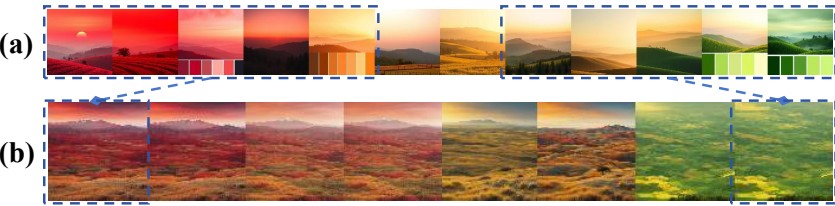

*a beautiful landscape*

Figure 19: Time-varying user preferences and corresponding generated results. (a) Illustrates the evolution of a user's preferred visual styles over time, shown by changes in color palettes and tone. (b) Images generated by PREFGEN conditioned on different temporal preference windows. The base prompt is "a beautiful landscape."

### D.7 TIME-VARYING USER PREFERENCES.

Figure 19 illustrates how PREFGEN adapts to time-varying user preferences. In practice, long-term preference histories can be viewed as a sequence of short-term, relatively stable windows, within which users tend to exhibit consistent visual and aesthetic tendencies. In Figure 19(a), we visualize the temporal evolution of a user's visual style preferences, where the dominant color palettes and overall tonal characteristics gradually change over time. Figure 19(b) presents the images generated by PREFGEN when conditioning on different temporal windows of the same user's preference history, while keeping the base textual prompt fixed as "a beautiful landscape". The results show that the model can effectively track and adapt to these temporal preference shifts, producing images whose color tones, atmosphere, and overall styles align with the user's most recent preference trends. These observations demonstrate that PREFGEN can model dynamic, time-varying personalization, rather than being limited to static preference settings.

### D.8 EXTENDED HUMAN EVALUATION ON PROLIFIC

To increase the diversity and statistical reliability of human judgments, and to provide a complementary validation of our results, we conducted a larger-scale user study on the Prolific platform (Prolific Team, 2014) with 50 participants and 900 pairwise comparisons. This study was designed to broaden the coverage of evaluators and to strengthen the robustness of the subjective assessment. Notably, compared to expert annotators, crowd participants are more likely to rely on salient surface-level cues such as color palettes and style patterns, often making rapid judgments based on first impressions, which provides a more realistic proxy for general user behavior. As shown in Table 14, our method is consistently preferred over all competing baselines in this crowd-based evaluation. We observe preference trends that are highly consistent with the expert study, while the preference margins become even more pronounced in the larger-scale setting (e.g., 97.33% against ViPer and over 88% against other methods). These results indicate that the observed improvements are robust across different evaluator populations and are not artifacts of a specific group of annotators, providing stronger evidence for the effectiveness of our approach in subjective aesthetic evaluation.

Table 14: Pairwise human preference results from the large-scale Prolific user study.

| Comparison | Winning rate (%) |
|---|---|
| Ours vs. Bagel | 78.67 |
| Ours vs. EasyRef | 88.00 |
| Ours vs. InstantStyle | 90.67 |
| Ours vs. IP-Adapter | 88.67 |
| Ours vs. StyleAligned | 88.67 |
| Ours vs. ViPer | 97.33 |

## D.9 SEMANTIC PREFERENCES

To evaluate whether our method captures semantic preferences rather than exhibiting stylistic bias, we design an experiment using both real and synthetic preference histories in Figure 20. Specifically, we construct user preference histories from (a–b) ImageNet images (Deng et al., 2009), where preference pairs are formed across different fine-grained dog categories, and (c) synthetic preference images generated by FLUX (Black Forest Labs, 2024a). During generation, we fix the text prompt to the same input for all cases, "a photo of a dog." For the synthetic histories in (c), we additionally normalize background and color distributions.

Across both real ImageNet-based and FLUX-generated preference histories, PrefGen consistently generates dogs that align with users' preferred fine-grained semantic categories. These results demonstrate that PrefGen captures semantic-level user preferences, rather than relying on superficial color or stylistic cues.

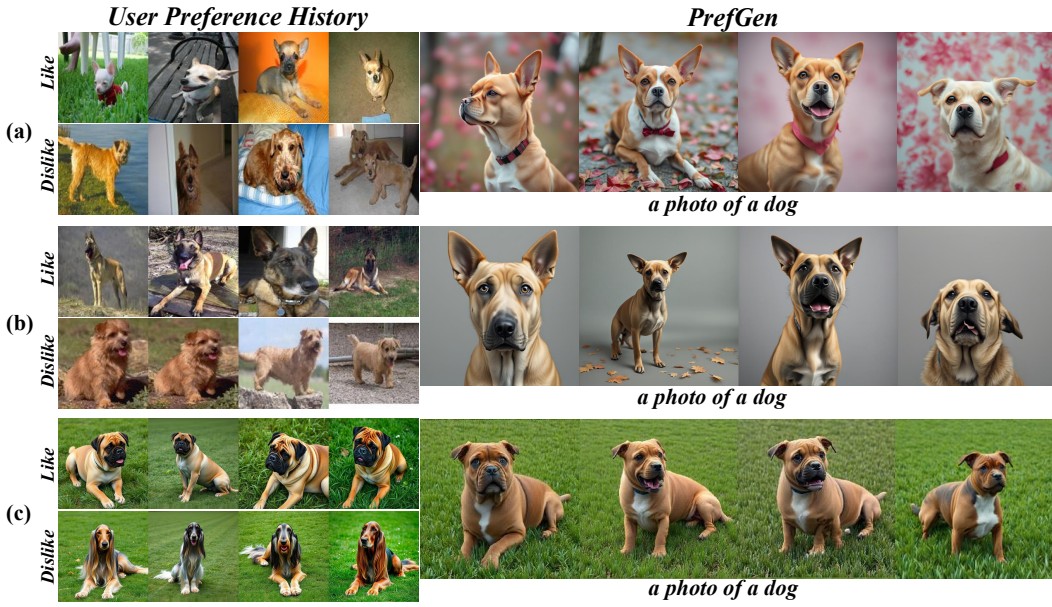

Figure 20: Semantic preference modeling with real and synthetic user preference histories. (a–b) User preference histories constructed from ILSVRC2012 ImageNet (Deng et al., 2009), where positive and negative examples are sampled from different fine-grained categories. (c) Synthetic preference histories generated by Flux.1-dev (Black Forest Labs, 2024a). PREFGEN successfully generates dog images that match users' fine-grained semantic preferences (e.g., breed type and facial morphology), demonstrating its ability to model semantic preferences beyond superficial style or color patterns.

## E    THE USE OF LARGE LANGUAGE MODELS(LLMs)

In this work, large language models (LLMs) are utilized solely as tools to assist with manuscript preparation. Specifically, they are employed for improving the clarity, coherence, and fluency of the text, as well as for checking grammar, spelling, and stylistic consistency. The models are not used to generate original research content, analyses, or results, and all scientific ideas, experimental designs, and conclusions are entirely the authors' own.

