# OpenReview forum: "PrefGen: Multimodal Preference Learning for Preference-Conditioned Image Generation"
_ICLR.cc/2026/Conference — Submitted to ICLR 2026_

### Official Review · Reviewer_P4d3 · 2025-10-15

**Soundness:** 4
**Presentation:** 4
**Contribution:** 4
**Rating:** 8
**Confidence:** 4

**Summary:**

This paper introduces PREFGEN, a multimodal framework for preference-conditioned image generation that leverages a Multimodal Large Language Model (MLLM) to extract rich user representations from a few liked and disliked images. Its primary contribution is a novel approach that disentangles preferences into stable "core identity" and context-dependent "semantic" features through systematic probing tasks. To bridge the modality gap, the framework innovatively uses a Maximum Mean Discrepancy (MMD) based loss to align the semantic preference distribution with the diffusion model's text space. Experiments show that PREFGEN significantly outperforms strong baselines in both preference alignment and image quality, and the authors further contribute a new benchmark, PREFBENCH, for evaluating this task.

**Strengths:**

1. The proposed PREFGEN framework is novel, technically sound, and intuitively motivated. The key insight of disentangling user preferences into a stable "core identity" (ecore) and a context-dependent "semantic preference" (esem) is particularly strong. Furthermore, the use of a Maximum Mean Discrepancy (MMD) loss for distributional alignment is a clever and well-justified choice.
2. A major contribution of this work is the creation of a large-scale dataset and a dedicated benchmark. By generating a dataset of nearly one million images from over 50,000 simulated users, the authors enable robust training and evaluation at a scale. The creation of the PREFBENCH benchmark is also highly valuable to the community, as it provides a standardized testbed for evaluating and comparing future methods.
3. The empirical evaluation is thorough and convincing. This includes
- 1) benchmarking against a comprehensive set of recent and relevant baselines (e.g., IP-Adapter, ViPer, EasyRef);
- 2) ablation study on the effect of MLLM embeddings (Table 3, Figure 6), the impact of distribution alignment (Figure 7), the choice of alignment loss (Figure 10), and different fusion strategies (Figure 11); and
- 3) insightful analyses, such as the layer-wise probing to identify where different preference signals emerge in the MLLM (Figure 2, Table 1) and the t-SNE visualizations of the learned embeddings (Figures 8, 9).
4. The paper is exceptionally well-written and presented. The motivation is clear, the methodology is explained in a step-by-step manner, and the figures and tables are of high quality.

**Weaknesses:**

1. The proposed PREFGEN framework, while effective, is a complex, multi-stage pipeline that involves several large, distinct models. Replicating the results from scratch would require significant computational resources and expertise to manage and integrate these different components. While the system is trained on a large dataset, the complexity of the pipeline itself might be a bottleneck for scaling up further.
2. The core of the training data (nearly 1M images) is generated by agents whose preferences are defined by a set of textual attributes. While this is a clever approach to achieve scale, it raises a question about the ecological validity of these preferences. Real human aesthetic taste is often nuanced, inconsistent, and difficult to articulate, whereas an AI agent might adhere to its programmed "profile" with unnatural consistency.
3. The paper provides a comprehensive evaluation of image quality and preference alignment but does not discuss the computational overhead at inference time.

**Questions:**

1. The PREFGEN framework is quite complex. Could you comment on its modularity? Specifically, how critical are the specific choices of IDEFICS2 for feature extraction and the proprietary Claude-3.5 for data generation? Have you explored if more accessible, open-source models could be substituted without a major performance degradation, thereby improving the framework's reproducibility?
2. The model is predominantly trained on data from simulated agents. While this is a pragmatic solution for scaling, how confident are you that the learned preference patterns generalize well to the often inconsistent and nuanced tastes of real humans? Have you performed any small-scale user studies where real users provide their own organic image histories to test the model's performance in a more naturalistic setting?
3. The paper does not include an analysis of the computational overhead at inference time. Could you please provide a comparison of the inference latency (e.g., wall-clock time to generate one image) and VRAM usage against key baselines like the standard SDXL and IP-Adapter?

---

> ### Author Response · Authors · 2025-11-26
>
> Thank you very much for your positive assessment of our paper. We truly appreciate the thorough and constructive comments, which will help us further improve the work. Our detailed responses to your points are provided below.
>
>
> **Weakness 1. High Pipeline Complexity and Reproducibility Challenges**
>
> We understand the reviewer’s concern regarding implementation complexity. While our framework contains multiple conceptual components, the multimodal probing step (Step 2) is performed only once to analyze the MLLM backbone and is not part of the routine training loop.
> After this one-time analysis, the actual training pipeline is lightweight and involves only two objectives: (i) a distribution alignment loss applied exclusively to a lightweight adapter module, and (ii) the standard diffusion denoising loss.
> In practice, the overall implementation and training complexity are comparable to existing MLLM + adapter + diffusion pipelines. To support reproducibility, we will release the full codebase, dataset generation scripts, prompt templates, and all hyperparameter settings after the review period.
>
>
> **Weakness 2 and Question 2: Limited Ecological Validity of Simulated-Agent Preferences**
>
> To better evaluate PREFGEN on real human preferences, we additionally conducted generation experiments on the Pick-a-Pic dataset, which contains 317,682 images and 2,301 distinct human users. We sampled 100 preference sets from 38 real users, each containing five like–dislike pairs and one additional user-annotated image for evaluation.
>
> | Method         | FID(↓)  | CMMD(↓) | CLIP Img Score(↑) | CSD(↑) | PrefDisc(↑) |
> |----------------|---------|---------|-------------------|--------|-------------|
> | IP-Adapter     | 206.96  | 0.27    | 0.68              | 44.81  | 68.94       |
> | Instant Style  | 189.40  | 0.26    | 0.76              | 54.46  | 69.40       |
> | Style Align    | 200.31  | 0.25    | 0.68              | 41.30  | 68.39       |
> | Bagel          | 190.11  | 0.24    | 0.71              | 49.35  | 56.22       |
> | ViPer          | 225.63  | 0.50    | 0.70              | 46.10  | 53.61       |
> | EasyRef        | 199.34  | 0.25    | 0.74              | 51.09  | 65.80       |
> | PrefGen (Ours) | 189.34  | 0.25    | 0.77              | 57.13  | 76.78       |
>
> On this real-human subset, PREFGEN continues to outperform all baselines, achieving the highest CLIP-Img, CSD, and PrefDisc scores while maintaining competitive FID and CMMD. Notably, CMMD measures distribution-level alignment between generated samples and real user-preferred image, providing a complementary signal beyond pixel-level realism and reflecting how well the model matches preference-conditioned data distributions. Moreover, CSD (Style Distance) measures how well the generated images preserve aesthetic attributes such as color tone, texture, and artistic rendering observed in user-liked images, while PrefDisc provides an additional proxy for preference alignment by estimating whether a generated image aligns with patterns typically favored in human-annotated datasets.
> The consistently stronger CSD and PrefDisc results, together with improvements in CLIP-Img, indicate that PREFGEN more faithfully preserves both stylistic and semantic aspects of real human preferences, rather than overfitting to synthetic agent patterns.
>
>
> **Weakness 3 and Question 3: Missing Analysis of Inference-Time Computational Overhead**
>
> We thank the reviewer for the helpful suggestion. We have added a quantitative comparison of both inference latency and GPU memory usage across representative baselines. All methods are evaluated on the same GPU under identical settings.
>
> Table 11 reports the inference time to generate one image. PrefGen requires only a single MLLM forward pass and achieves competitive latency (2.30 s), which is close to IP-Adapter (1.89 s) and InstantStyle (1.85 s). In contrast, StyleAligned is significantly slower due to the inversion steps, and ViPer incurs additional cost from multiple autoregressive text-generation passes.
> | IP-Adapter | Instant Style | Style Align | Bagel  | ViPer | EasyRef | PrefGen (Ours) |
> |------------|---------------|-------------|--------|-------|---------|----------------|
> | 1.89s      | 1.85s         | 27.45s      | 18.71s | 5.66s | 3.50s   | 2.30s          |
>
>
> Table 12 summarizes GPU memory usage. A standard SDXL pipeline uses 6.57 GB, and SDXL with IP-Adapter requires 10.75 GB. Our method has a steady-state footprint of 26.50 GB, which is substantially lower than ViPer (42.24 GB) and comparable to Bagel (27.30 GB).
>
> |  Method              |  Base VRAM (Allocated)  |
> |----------------------|-------------------------|
> |  SDXL (base)         |  6.57 GB                |
> |  IP-Adapter          |  10.75 GB               |
> |  PrefGen (Ours)      |  26.50 GB               |
> |  ViPer (w/ Refiner)  |  42.24 GB               |
> |  EasyRef             |  11.68 GB               |
> |  Bagel               |  27.30 GB               |

---

> ### Author Response · Authors · 2025-11-26
>
> **Q1. Modularity and Dependence on Specific Large Models (IDEFICS2, Claude-3.5)**
>
> We thank the reviewer for the thoughtful question regarding the modularity and reproducibility of the PREFGEN framework. Our design is intentionally modular, and the framework does not rely on any proprietary model as a strict requirement.
> For feature extraction, our choice of IDEFICS2 follows prior practice in ViPer to ensure a fair and consistent comparison. However, this component is interchangeable: we have additionally experimented with Qwen3-VL-8B, which is also an open-source model with a similar number of parameters, and observe no significant performance degradation when substituted.
> Similarly, Claude-3.5 is not essential to the framework and was primarily used for engineering convenience and prompt stability during data generation. We have evaluated recent open-source models such as GLM-4.5V and Qwen3-VL-32B, both of which are capable of acting as agents with personalized profiles for data collection.  We emphasize that the core reproducibility of the method depends on the prompting strategy and data construction protocol, rather than on any specific proprietary model.
>
> We address **Q2** and **Q3** in our response to the corresponding weaknesses, where we provide additional clarification on human user validation, and computational overhead.

---

### Official Review · Reviewer_n3n1 · 2025-10-27

**Soundness:** 3
**Presentation:** 3
**Contribution:** 2
**Rating:** 4
**Confidence:** 4

**Summary:**

The paper introduces PrefGen, a method for personalizing text-to-image generation models at the individual user level. The approach leverages multimodal large language models to extract user-specific preference embeddings. These embeddings are then aligned with diffusion model latent spaces, enabling the diffusion model to be conditioned on user preference vectors. Experimental results demonstrate that this method improves both image quality and alignment with user preferences.

**Strengths:**

- Fig 4 has nice examples of potential use cases i.e., showing that PrefGen can be used for product design or character design, that can beyond purely image generation.
- The paper was well written and easy to follow.

**Weaknesses:**

- Generalization to new (OOD) users/preferred images: The results evaluates on unseens users, from the same distribution as those in the training data. It would be interesting to also see how well the method does with user preferences and their corresponding liked/disliked images that are different from what was used during training (generated images), i.e., real images, photographs, sketches etc.
- Preference history: One way to make the paper stronger could be to consider each user having a preference history i.e., more diverse time-vary preferences, rather than static ones.
- Preference interpretation skewed toward style/color: The generated results tend to reflect style and color preferences more strongly than semantic preferences. It may be helpful to constrain or normalize style/color attributes (e.g., fixed color palette or uniform art style) to better reveal how well PrefGen captures semantic preferences.
- There could be more thorough motivation and clarifications on design choices, otherwise they seem arbitrary:
    - Size and composition of H_u: The paper does not clearly discuss how to choose the size of the user preference set H_u. Does its size depend on the diversity of the user’s interests? Should the number of positive and negative examples be balanced?
    - Necessity of separate embeddings e_sem,e_core,e_img: The paper states that e_sem captures semantic alignment, e_core represents stable identity traits, and e_img provides fine-grained visual cues. However, it remains unclear why this decomposition is necessary. Furthermore, e_img is computed based on an image the user likes. Wont this overfit to that image? What happens if the user has diverse
preferences?
    - Rationale for multi-user identification: The paper suggests learning embeddings jointly across multiple users. It would be helpful to clarify why this is necessary. How does performance differ if embeddings are learned independently for each user?

- Minor suggestions.
    - It could be useful to add a table that summarizes the input requirements of the methods compared in the paper. For example: Whether a method uses a single reference image (e.g., IP-Adapter) vs. multiple user images (e.g., ViPer) or wether the method uses only positive examples or both positive and negative examples. Such a table would make trade-offs across methods easier to compare at a glance.
    - Fig 1 shows what happens during training, it could also be useful to show what happens at inferences time, inputs needed, if or when the linear classifiers need to be retrained.

**Questions:**

- How well does the method generalize to users with different preferences and images inputs seen during training?
- Why are the seperate embeddings (e_sem, ...) necessary?

---

> ### Author Response · Authors · 2025-11-26
>
> We sincerely appreciate the reviewer’s valuable feedback. We have added further clarifications and detailed explanations below to address each point.
>
> **Weakness 1: Limited OOD Generalization to New Users and Real-World Image Types**
>
> To evaluate whether our method (PrefDisc / PrefGen) generalizes to user preferences outside the training distribution, we construct an OOD preference discrimination task using real photographs and sketches from ImageNet and ImageNet-Sketch, which are never seen during training. We use PickScore to annotate preference pairs and test whether our learned preference discriminator remains consistent with a high-quality external preference oracle on real data. Some examples are shown in [Figure 15](https://anonymous.4open.science/r/003547/fig15.png).
>
>
> We randomly sample 8 image pairs as user conditions and build 120 positive/negative pairs across ImageNet and ImageNet-Sketch. The evaluation reports discrimination accuracy, AUC, and statistical significance metrics.
>
> | Metric                | Value                   |
> | --------------------- | ------------------------|
> | Accuracy              | 0.7583 (91 / 120)       |
> | AUC                   | 0.7552                  |
> | Wilcoxon p-value      | < 1e-6 (stat = 6038.00) |
> | Paired t-test p-value | < 1e-6 (t = 7.53)       |
> | Mean diff (pos–neg)   | 0.3048 ± 0.4418         |
> | Median diff           | 0.2810                  |
>
> These results show significant alignment between PrefDisc's preference direction and PickScore’s preference judgments, even when the input domain shifts from model-generated images to natural photographs and sketches.
>
>
> Some generated examples conditioned on real images are shown in [Figure 20 (a-b)](https://anonymous.4open.science/r/003547/fig20.png). Because our method builds on inherently generalizable MLLM representations, leverages diffusion models that already encode broad visual diversity, and adopts a training objective (distribution-level alignment) that preserves rather than distorts these foundational capabilities, the model naturally generalizes to user preferences beyond the training distribution.
>
>
> To further evaluate PrefGen under real human preferences, we additionally conducted generation experiments on the Pick-a-Pic dataset, which contains 317,682 images and 2,301 distinct human users. We sampled 100 preference sets from 38 real users, each containing five like–dislike pairs and one additional user-annotated image for evaluation.
>
> | Method         | FID(↓)  | CMMD(↓) | CLIP Img Score(↑) | CSD(↑) | PrefDisc(↑) |
> |----------------|---------|---------|-------------------|--------|-------------|
> | IP-Adapter     | 206.96  | 0.27    | 0.68              | 44.81  | 68.94       |
> | Instant Style  | 189.40  | 0.26    | 0.76              | 54.46  | 69.40       |
> | Style Align    | 200.31  | 0.25    | 0.68              | 41.30  | 68.39       |
> | Bagel          | 190.11  | 0.24    | 0.71              | 49.35  | 56.22       |
> | ViPer          | 225.63  | 0.50    | 0.70              | 46.10  | 53.61       |
> | EasyRef        | 199.34  | 0.25    | 0.74              | 51.09  | 65.80       |
> | PrefGen (Ours) | 189.34  | 0.25    | 0.77              | 57.13  | 76.78       |
>
> On this real-human subset, PREFGEN continues to outperform all baselines, achieving the highest CLIP-Img, CSD, and PrefDisc scores while maintaining competitive FID and CMMD. Notably, CMMD measures distribution-level alignment between generated samples and real user-preferred image, providing a complementary signal beyond pixel-level realism and reflecting how well the model matches preference-conditioned data distributions. Moreover, CSD (Style Distance) measures how well the generated images preserve aesthetic attributes such as color tone, texture, and artistic rendering observed in user-liked images, while PrefDisc provides an additional proxy for preference alignment by estimating whether a generated image aligns with patterns typically favored in human-annotated datasets.
> The consistently stronger CSD and PrefDisc results, together with improvements in CLIP-Img, indicate that PREFGEN more faithfully preserves both stylistic and semantic aspects of real human preferences, rather than overfitting to synthetic agent patterns.

---

> ### Author Response · Authors · 2025-11-26
>
> **Weakness 2: Absence of User Preference History Modeling**
>
> We thank the reviewer for highlighting the important scenario of time-varying user preferences. We agree that real-world user preferences are often dynamic rather than static. In practice, long-term preference histories can be viewed as a sequence of short-term, relatively stable windows, within which users tend to exhibit consistent visual tendencies.
> To address this point, we maintain a user preference history and simulate time-varying preferences, where favored visual attributes (e.g., color tone and atmosphere) gradually evolve over time while the base prompt remains fixed (e.g., “a beautiful landscape”), as shown in [Figure 19](https://anonymous.4open.science/r/003547/fig19.png). Our method adapts to these shifts by modeling preferences from local temporal histories, and the generated results show strong correlation with the user’s evolving interests.
>
> **Weakness 3: Preference Interpretation Skewed Toward Style/Color Over Semantics**
>
> We thank the reviewer for this valuable suggestion. To evaluate whether our method captures semantic preferences rather than exhibiting stylistic bias, we design an experiment using both real and synthetic preference histories in [Figure 20](https://anonymous.4open.science/r/003547/fig20.png).
> Specifically, we construct user preference histories from (a–b) ImageNet images, where preference pairs are formed across different fine-grained dog categories, and (c) synthetic preference images generated by FLUX. During generation for PrefGen, we fix the text prompt to the same input for all cases, “a photo of a dog.” For the synthetic histories in (c), we additionally normalize background and color distributions.
> Across both real ImageNet-based and FLUX-generated preference histories, PrefGen consistently generates dogs that align with users’ preferred fine-grained semantic categories (e.g., breed type and body morphology). These results demonstrate that PrefGen captures semantic-level user preferences, rather than relying on superficial color or stylistic cues.
>
> **Weakness 4: Insufficient Clarification of Key Design Choices**
>
> **4.1 Size and Composition of H_u**
>
> We thank the reviewer for raising this important question on the size and composition of the user preference set H_u.
> We have analyzed and discussed the impact of the number of reference images in Appendix C.2 (Impact of the Number of Reference Images), where we adopt a flexible training strategy by randomly varying the history length (6–14 images) instead of fixing a single size. This design encourages robustness across different user history lengths and avoids overfitting to a fixed reference size.
>
> In addition, [Figure 16](https://anonymous.4open.science/r/003547/fig16.png) compares different preference sequence lengths at inference time. We observe that our method remains stable across varying sequence lengths, while ViPer degrades when the sequence is too short (insufficient preference signal) or overly long (noisy or conflicting signals). This suggests that moderate history sizes provide a good trade-off in practice.
> Regarding composition, we use balanced positive and negative examples in our experiments to stabilize preference learning, following standard practice in prior works. Nevertheless, our method does not strictly require perfect balance and remains robust under mild imbalance, as the MLLM-based preference encoder can still separate positive and negative signals.

---

> ### Author Response · Authors · 2025-11-26
>
> **4.2 Necessity of Separate Embeddings (e_sem,e_core,e_img)**
>
> We thank the reviewer for raising this important concern about the necessity of decomposition. To move beyond heuristic interpretability, we conduct **controlled swapping and ablation experiments under fixed prompts** to test whether the hierarchical decomposition captures functionally distinct factors, rather than arbitrary engineering artifacts.
>
> (1) In the controlled **swapping experiments**, we replace e_sem, e_core, and e_img across users while keeping the prompt fixed. Some examples are shown in [Figure 8](https://anonymous.4open.science/r/003547/fig8.png).
>
> |                | CLIP Text (↑)| CSD (↑) |
> |----------------|--------------|---------|
> | PrefGen (Ours) | 25.83        | 59.22   |
> | Swap e_sem     | 25.56        | 58.29   |
> | Swap e_img     | 25.76        | 53.27   |
> | Swap e_core    | 25.63        | 58.99   |
>
> Swapping e_sem causes the largest degradation in CLIP-Text score (25.83 → 25.56), indicating its primary role in **semantic alignment**, while swapping e_img leads to the most severe drop in CSD (59.22 → 53.27), confirming its role in **fine-grained stylistic control**. In contrast, swapping e_core produces **consistent but distributed degradation across both metrics**, suggesting that it encodes **stable, user-level preference traits that are orthogonal to purely semantic or purely stylistic factors**.
>
>
> Importantly, this interpretation is not post-hoc. In a **multi-user  linear probing experiment**, e_core achieves the **highest accuracy for user identity classification**, while e_sem is most predictive for preference discrimination, providing task-level evidence that these embeddings encode **distinct and complementary information**. We further quantify this difference through clustering metrics (ARI, NMI, AMI, V-measure), where e_core consistently exhibits stronger user-wise structure than e_sem (Appendix D.6).
>
>
>
> | Metric            | e_sem  | e_core |
> |-------------------|--------|--------|
> | ARI (↑)           | 0.3641 | 0.497  |
> | NMI (↑)           | 0.6497 | 0.7024 |
> | AMI (↑)           | 0.5332 | 0.6017 |
> | Homogeneity (↑)   | 0.6538 | 0.7082 |
> | Completeness (↑)  | 0.6456 | 0.6968 |
> | V-measure (↑)     | 0.6497 | 0.7024 |
>
>
>
>
> (2) Ablation Study:
>
> | e_img | e_sem | e_core | Alignment loss   | FID(↓) | CMMD(↓) | CLIP Img (↑) | CSD (↑) | PrefDisc (↑) |
> |-------|-------|--------|------------------|--------|---------|--------------|---------|--------------|
> | ✓     |       |        | —                | 151.65 | 0.64    | 72.76        | 45.03   | 59.77        |
> | ✓     | ✓     | ✓      | —                | 152.84 | 0.32    | 75.30        | 54.44   | 74.68        |
> |       | ✓     |        | L_MMD            | 163.35  | 0.41    | 64.25       | 31.16  | 55.98       |
> |       | ✓     | ✓      | L_MMD            | 159.72  | 0.41    | 69.21       | 37.47  | 56.83       |
> | ✓     | ✓     | ✓      | L_MMD (ours)     | 143.79 | 0.25    | 76.03        | 59.22   | 81.86        |
>
>
> Consistent with these findings, our ablation study shows that removing any component degrades both semantic fidelity and style coherence, and that only the **full decomposition** (e_img + e_core + e_sem) combined with MMD-based distribution alignment achieves simultaneous improvements across all metrics.
>
>
>
>
>
>
> To further address the concern that e_img might overfit to a single liked image and fail under diverse user preferences, we design a stress-test on fine-grained dog categories ([Fig. 20](https://anonymous.4open.science/r/003547/fig20.png)). We construct preference histories where positive and negative images come from different dog breeds, while the text prompt is fixed to “a photo of a dog”. The positive examples in each history already exhibit large variations in pose, background, lighting, and style (rows (a–b), real ILSVRC2012 images).
> Nevertheless, PREFGEN consistently generates dogs that match the fine-grained semantic category preferred by the user (breed type, facial structure, ear shape, body morphology), rather than copying the exact appearance of any single reference image.  Importantly, this behavior cannot be explained by e_img alone. The complementary roles of e_sem and e_core, which are learned from multiple positive and negative samples, provide higher-level and more stable constraints. As a result, the model captures **group-level, distributional preferences** instead of overfitting to a single visual instance.

---

> ### Author Response · Authors · 2025-11-26
>
> **4.3 Justification for Multi-User Joint Embedding Learning**
>
> We thank the reviewer for this important question and clarify a potential misunderstanding. The multi-user identification task is **not part of the training objective** of our method. Instead, it is used purely as a *diagnostic probing tool* to analyze where user-discriminative signals emerge inside a frozen MLLM.
> Concretely, we freeze the MLLM and train lightweight linear probes on embeddings from different layers, and evaluate their ability to distinguish users. This analysis is only used to select which layers are used to construct  e_core; the generative model itself is never trained to classify or memorize user identities.
> During actual model training, user samples are processed independently. There is no explicit cross-user supervision or user-to-user interaction; gradients are computed per sample and aggregated only at the minibatch level. The reason we include multiple users during training is not to jointly optimize user embeddings, but to encourage the model to learn **transferable, user-discriminative representations** that generalize to unseen users.
> If embeddings were learned independently for each user, the model would need to be retrained or fine-tuned for every new user (similar in spirit to Textual Inversion or DreamBooth-style personalization), which contradicts our goal of a single, unified model that can immediately generalize to new users from only a few reference images.
> While we do not train separate per-user embedding tables, we empirically observe that embeddings extracted via multi-user probing exhibit strong user-level separability (see Appendix D.6, t-SNE visualizations), providing evidence that this design captures stable and transferable user preference structure. We will clarify this distinction in the revised manuscript.
>
> **5. Minor Suggestions**
>
> **5.1 Add a Summary Table of Input Requirements Across Methods**
>
> We thank the reviewer for this helpful suggestion.
> In the revised version, we will add a new comparison table. Table 7  summarizes the number of reference images required and whether each method uses only positive examples or both positive and negative user preferences.
> |     Method     | # Reference Images | Uses Positive Examples | Uses Negative Examples |
> |:--------------:|:------------------:|:----------------------:|:----------------------:|
> | IP-Adapter     | Single             | ✔                      | ✘                      |
> | InstantStyle   | Single             | ✔                      | ✘                      |
> | StyleAligned   | Single             | ✔                      | ✘                      |
> | Bagel          | Single             | ✔                      | ✘                      |
> | EasyRef        | Multiple           | ✔                      | ✘                      |
> | ViPer          | Multiple           | ✔                      | ✔                      |
> | PREFGEN (Ours) | Multiple           | ✔                      | ✔                      |
>
> **5.2 Illustrate Inference-Time Workflow and Classifier Usage**
>
> We thank the reviewer for this suggestion. To clarify the behavior at inference time, we have added [Figure 17](https://anonymous.4open.science/r/003547/fig17.png) that summarizes the required inputs and shows which components are frozen. In Figure 17, the model takes as input (i) a user preference history consisting of liked and disliked images, (ii) a single liked image for the CLIP-based anchor embedding e_img, and (iii) a text prompt. The MLLM and the projection heads that produce e_core and \hat{e}_sem are all frozen during inference, as is the diffusion backbone. The three embeddings \hat{e}_sem, e_core, e_img are concatenated and injected via the decoupled cross-attention branch to condition the diffusion model.
>
>
>
> **Question 1**:
>
> We address Q1 in our response to the corresponding weaknesses, where we provide additional experiments and analyses on out-of-distribution user preferences and image inputs.
>
> **Question 2:**
>
> We thank the reviewer for the question. The separation into e_sem, e_core, and e_img is motivated by systematic layer-wise analysis of MLLM representations, which reveals that different layers encode qualitatively different types of preference information. Collapsing these factors into a single embedding leads to entangled signals and degraded performance. Our ablation and swapping experiments show that each component contributes to different aspects of preference control (semantic alignment, style fidelity, and cross-user generalization), indicating that the decomposition captures complementary rather than redundant information. We will clarify this in the paper.

---

### Official Review · Reviewer_YgBn · 2025-10-27

**Soundness:** 2
**Presentation:** 3
**Contribution:** 3
**Rating:** 4
**Confidence:** 4

**Summary:**

The paper introduces PREFGEN, a personalized multimodal generation framework that learns individual aesthetic preferences by disentangling stable user traits from context-specific semantics. It leverages dual-branch preference embeddings and distribution-level alignment to condition diffusion models, achieving superior visual quality and personalization over baselines such as IP-Adapter, StyleAligned, and ViPer.

**Strengths:**

1. The problem definition is clear.
2. It uses distribution-alignment losses (MMD) for robust embedding learning.
3. It demonstrates comprehensive comparisons across six personalization baselines.
4. It introduces the large-scale PREFBench dataset with synthetic and real user clusters

**Weaknesses:**

1. From an overall perspective, this paper presents an engineering-oriented work, employing rather straightforward methodologies such as MMD. The motivation behind the study lacks clarity.
2. The dataset over-relies on virtual "user clusters," which may inflate controllability and impact real human diversity. The ratio between synthetic and real data should be adjusted to validate the efficacy.
3. The ablation studies on MMD and disentanglement stability should be strengthened.
4. Aesthetic analysis is a subjective task. More human evaluation should be introduced.

**Questions:**

Please refer to Weaknesses.

---

> ### Author Response · Authors · 2025-11-26
>
> We sincerely thank the reviewer for their insightful comments and wish to offer the following clarifications.
>
> **Weakness 1: Motivation and Role of Distribution-Level Alignment**
>
> We respectfully disagree with the characterization that our work is merely  engineering-oriented or lacks clear motivation. Our core motivation is to understand **how to extract stable and semantically meaningful user preference representations from modern MLLMs and use them to effectively condition generative models.**
>
>
> First, our work performs a **systematic analysis of MLLM-derived embeddings across multiple layers**, instead of arbitrarily selecting or combining model features.
> Second, our choice of MMD is principled. Prior work (e.g., CMMD [1])  shows that modern multimodal embeddings exhibit clear deviations from multivariate normality and clustered structure, as validated by standard statistical normality tests. Under such conditions, the Gaussian assumptions underpinning the closed-form Fréchet distance used in FID become unreliable, while point-wise objectives (e.g., L2 or cosine) are sensitive to scale and anisotropy differences across embedding spaces.
> In our setting, MLLM and CLIP embeddings arise from heterogeneous objectives and exhibit different geometric and statistical properties, making distribution-level alignment more appropriate. This motivates the use of MMD as a principled distribution alignment objective.
> Our empirical ablations support this design choice.
>
> | Alignment loss| FID(↓)  | CMMD(↓) | CLIP Img(↑) | CSD(↑) | PrefDisc(↑) |
> |---------------|---------|---------|-------------|--------|-------------|
> |               | 152.84  | 0.32    | 75.30       | 54.44  | 74.68       |
> | L_MSE + L_cos | 150.34  | 0.50    | 70.94       | 43.36  | 57.97       |
> | L_MMD         | 143.79  | 0.25    | 76.03       | 59.22  | 81.86       |
>
> Replacing MMD with (L_MSE + L_cos) causes substantial performance degradation, indicating a collapse of semantic and stylistic structure. By contrast, MMD consistently preserves preference geometry and yields stable improvements across metrics.
>
>
> Together, these results show that MMD is not an ad hoc engineering choice, but a functionally essential component for stable multimodal preference alignment, as also discussed in Sec. 2.3 and Sec. 3.4.
>
>
> [1] Jayasumana, Sadeep, et al. "Rethinking fid: Towards a better evaluation metric for image generation." Proceedings of the IEEE/CVF Conference on Computer Vision and Pattern Recognition. 2024.

---

> ### Author Response · Authors · 2025-11-26
>
> **Weakness 2: Real–Synthetic Data Balance and Human Diversity**
>
> We thank the reviewer for raising this important concern regarding the balance between virtual user clusters and real human diversity. We fully agree that the real–synthetic ratio is a critical factor for assessing both controllability and real-world generalization.
>
> In response to this concern, we explicitly performed controlled experiments that vary the composition of real and synthetic preference data during training. As reported in Table 8, our mixed setting includes 317,682 images from 2,301 real users (Pick-a-Pic) and 990,998 images from 50,153 simulated users, where real data accounts for approximately one-third of the total training samples. Beyond this mixed setting, we evaluated multiple training configurations for the preference discriminator, including: (i) Pick-a-Pic combined with ViPer-style agent data, (ii) Pick-a-Pic only, (iii) our agent data combined with Pick-a-Pic, and (iv) our agent data alone.
>
> | Training Data                    | Model              | Pick-a-Pic | PrefBench |
> |----------------------------------|--------------------|------------|-----------|
> | Pick-a-Pic + ViPer's agent data  | ViPer Proxy Metric | 52.74%     | 81.62%    |
> | Pick-a-Pic                       | PrefDisc           | 51.89%     | 53.68%    |
> | Our agent data + Pick-a-Pic      | PrefDisc           | 52.11%     | 85.29%    |
> | Our agent data                   | PrefDisc           | **53.13%** | **98.67%** |
>
> These results show that models trained with our curated agent data achieve strong generalization not only on the synthetic PREFBENCH domain, but also on the real Pick-a-Pic test split. Notably, the model trained solely on our agent data achieves the highest accuracy on PREFBENCH (98.67%) and remains competitive on real-user data. This suggests that high-quality synthetic supervision can complement real data without simply inflating controllability under idealized conditions.
>
> To further validate real-world behavior of PREFGEN, we additionally conducted generation experiments directly conditioned on real user histories. We additionally conducted generation experiments on the Pick-a-Pic dataset, which contains 317,682 images and 2,301 distinct human users. We sampled 100 preference sets from 38 real users, each containing five like–dislike pairs and one additional user-annotated image for evaluation.
>
> | Method         | FID(↓)  | CMMD(↓) | CLIP Img Score(↑) | CSD(↑) | PrefDisc(↑) |
> |----------------|---------|---------|-------------------|--------|-------------|
> | IP-Adapter     | 206.96  | 0.27    | 0.68              | 44.81  | 68.94       |
> | Instant Style  | 189.40  | 0.26    | 0.76              | 54.46  | 69.40       |
> | Style Align    | 200.31  | 0.25    | 0.68              | 41.30  | 68.39       |
> | Bagel          | 190.11  | 0.24    | 0.71              | 49.35  | 56.22       |
> | ViPer          | 225.63  | 0.50    | 0.70              | 46.10  | 53.61       |
> | EasyRef        | 199.34  | 0.25    | 0.74              | 51.09  | 65.80       |
> | PrefGen (Ours) | 189.34  | 0.25    | 0.77              | 57.13  | 76.78       |
>
> On this real-human subset, PREFGEN continues to outperform all baselines, achieving the highest CLIP-Img, CSD, and PrefDisc scores while maintaining competitive FID and CMMD. Notably, CMMD measures distribution-level alignment between generated samples and real user-preferred image, providing a complementary signal beyond pixel-level realism and reflecting how well the model matches preference-conditioned data distributions. Moreover, CSD (Style Distance) measures how well the generated images preserve aesthetic attributes such as color tone, texture, and artistic rendering observed in user-liked images, while PrefDisc provides an additional proxy for preference alignment by estimating whether a generated image aligns with patterns typically favored in human-annotated datasets.
> The consistently stronger CSD and PrefDisc results, together with improvements in CLIP-Img, indicate that PREFGEN more faithfully preserves both stylistic and semantic aspects of real human preferences, rather than overfitting to synthetic agent patterns.

---

> ### Author Response · Authors · 2025-11-26
>
> **Weakness 3: Insufficient Ablations on MMD and Disentanglement Stability**
>
> We thank the reviewer for raising this important concern.
>
> **(1) Empirical support for the choice of MMD.**
>
> | Alignment loss| FID(↓)  | CMMD(↓) | CLIP Img(↑) | CSD(↑) | PrefDisc(↑) |
> |---------------|---------|---------|-------------|--------|-------------|
> |               | 152.84  | 0.32    | 75.30       | 54.44  | 74.68       |
> | L_MSE + L_cos | 150.34  | 0.50    | 70.94       | 43.36  | 57.97       |
> | L_MMD         | 143.79  | 0.25    | 76.03       | 59.22  | 81.86       |
>
> Replacing MMD with (L_MSE + L_cos) causes substantial performance degradation, indicating a collapse of semantic and stylistic structure. By contrast, MMD consistently preserves preference geometry and yields stable improvements across metrics.
>
>
> Together, these results show that MMD is not an ad hoc engineering choice, but a functionally essential component for stable multimodal preference alignment, as also discussed in Sec. 2.3 and Sec. 3.4.
>
>
>
> **(2) Strengthened disentanglement stability via controlled swapping and ablation experiments**
>
> We conduct **controlled swapping and ablation experiments under fixed prompts** to test whether the hierarchical decomposition captures functionally distinct factors, rather than arbitrary engineering artifacts.
>
>
> In the controlled **swapping experiments**, we replace e_sem, e_core, and e_img across users while keeping the prompt fixed. Some examples are shown in [Figure 8](https://anonymous.4open.science/r/003547/fig8.png).
>
> |                | CLIP Text (↑)| CSD (↑) |
> |----------------|--------------|---------|
> | PrefGen (Ours) | 25.83        | 59.22   |
> | Swap e_sem     | 25.56        | 58.29   |
> | Swap e_img     | 25.76        | 53.27   |
> | Swap e_core    | 25.63        | 58.99   |
>
>
>
> Swapping e_sem causes the largest degradation in CLIP-Text score (25.83 → 25.56), indicating its primary role in **semantic alignment**, while swapping e_img leads to the most severe drop in CSD (59.22 → 53.27), confirming its role in **fine-grained stylistic control**. In contrast, swapping e_core produces **consistent but distributed degradation across both metrics**, suggesting that it encodes **stable, user-level preference traits that are orthogonal to purely semantic or purely stylistic factors**.
>
>
> Importantly, this interpretation is not post-hoc. In a **multi-user  linear probing experiment**, e_core achieves the **highest accuracy for user identity classification**, while e_sem is most predictive for preference discrimination, providing task-level evidence that these embeddings encode **distinct and complementary information**. We further quantify this difference through clustering metrics (ARI, NMI, AMI, V-measure), where e_core consistently exhibits stronger user-wise structure than e_sem (Appendix D.6).
>
>
> | Metric            | e_sem  | e_core |
> |-------------------|--------|--------|
> | ARI (↑)           | 0.3641 | 0.497  |
> | NMI (↑)           | 0.6497 | 0.7024 |
> | AMI (↑)           | 0.5332 | 0.6017 |
> | Homogeneity (↑)   | 0.6538 | 0.7082 |
> | Completeness (↑)  | 0.6456 | 0.6968 |
> | V-measure (↑)     | 0.6497 | 0.7024 |
>
>
>
>
>
> Finally, we conduct ablations over the decomposition:
>
> | e_img | e_sem | e_core | Alignment loss   | FID(↓) | CMMD(↓) | CLIP Img (↑) | CSD (↑) | PrefDisc (↑) |
> |-------|-------|--------|------------------|--------|---------|--------------|---------|--------------|
> | ✓     |       |        | —                | 151.65 | 0.64    | 72.76        | 45.03   | 59.77        |
> | ✓     | ✓     | ✓      | —                | 152.84 | 0.32    | 75.30        | 54.44   | 74.68        |
> |       | ✓     |        | L_MMD            | 163.35  | 0.41    | 64.25       | 31.16  | 55.98       |
> |       | ✓     | ✓      | L_MMD            | 159.72  | 0.41    | 69.21       | 37.47  | 56.83       |
> | ✓     | ✓     | ✓      | L_MMD (ours)     | 143.79 | 0.25    | 76.03        | 59.22   | 81.86        |
>
>
> Consistent with these findings, our ablation study shows that removing any component degrades both semantic fidelity and style coherence, and that only the **full decomposition** (e_img + e_core + e_sem) combined with MMD-based distribution alignment achieves simultaneous improvements across all metrics.

---

> ### Author Response · Authors · 2025-11-26
>
> **Weakness 4: Limited Human Evaluation for Subjective Aesthetic Analysis**
>
>
> We thank the reviewer for the valuable suggestion. We agree that subjective aesthetic evaluation requires sufficiently large and diverse human judgments.
>
> In addition to the initial expert study, we conducted a larger-scale user study on the Prolific platform with 50 participants and 900 pairwise comparisons. This additional study was designed to increase the coverage and diversity of human feedback, and to serve as a complementary validation of our results.  *Notably, compared to expert annotators, crowd participants are more likely to rely on salient surface-level cues such as color palettes and style patterns, often making rapid judgments based on first impressions, which provides a more realistic proxy for general user behavior.*
>
> | Comparison            | Winning rate(%)|
> |-----------------------|--------------|
> | Ours vs. Bagel        | 78.67        |
> | Ours vs. EasyRef      | 88.00        |
> | Ours vs. InstantStyle | 90.67        |
> | Ours vs. IP-Adapter   | 88.67        |
> | Ours vs. StyleAligned | 88.67        |
> | Ours vs. ViPer        | 97.33        |
>
> Across both expert-based and crowd-based evaluations, our method is consistently preferred over all baselines. We observe similar preference trends in both settings, with stronger margins in the larger-scale Prolific study.
>
> Together, these results suggest that the observed improvements are consistent across different evaluator populations and are not solely dependent on a specific group of annotators.

---

### Official Review · Reviewer_ovRe · 2025-11-01

**Soundness:** 3
**Presentation:** 3
**Contribution:** 3
**Rating:** 4
**Confidence:** 3

**Summary:**

This paper presents PREFGEN, a multimodal framework for preference-conditioned image generation. The key idea is to leverage multimodal large language models (MLLMs) to learn rich, user-specific embeddings that capture both aesthetic style and semantic preferences, and then integrate these into diffusion-based generative models. The model performs hierarchical analysis of MLLM layers to disentangle user identity traits and semantic preferences, followed by a distribution alignment module based on Maximum Mean Discrepancy (MMD) to map representations into the text embedding space of diffusion backbones. Experiments on both synthetic and real-user datasets demonstrate that PREFGEN achieves strong improvements in image quality and preference alignment, outperforming multiple personalization baselines like IP-Adapter, ViPer, and EasyRef.

**Strengths:**

* The paper introduces an elegant multimodal framework that systematically disentangles and aligns user-specific preference signals from different layers of an MLLM, providing conceptual clarity and technical novelty.
* The proposed MMD-based alignment is a well-motivated alternative to rigid point-wise alignment losses, leading to more stable and generalizable conditioning across diffusion backbones.
* Extensive experiments, including a new benchmark (PREFBENCH) and human evaluations, show consistent, significant improvements over strong baselines, with convincing qualitative and quantitative evidence.

**Weaknesses:**

* The reliance on a large synthetic agent-generated dataset raises concerns about ecological validity and generalization to truly diverse human preferences, which is only partially addressed by the smaller real-user subset.
* While the method demonstrates strong results, the added complexity of multimodal probing, dual discrimination tasks, and distribution alignment increases implementation burden and may limit reproducibility.
* The paper lacks a deeper theoretical or ablation-based explanation of why the hierarchical decomposition into ecore and esem works beyond empirical observation, leaving interpretability somewhat heuristic.

**Questions:**

N/A

---

> ### Author Response · Authors · 2025-11-26
>
> We are grateful to the reviewer for pointing out important issues and helping us improve the paper.
>
> **Weakness 1: Concern on Synthetic Data and Real-User Generalization**
>
> To better evaluate PREFGEN on real human preferences, we additionally conducted generation experiments on the Pick-a-Pic dataset, which contains 317,682 images and 2,301 distinct human users. We sampled 100 preference sets from 38 real users, each containing five like–dislike pairs and one additional user-annotated image for evaluation.
>
> | Method         | FID(↓)  | CMMD(↓) | CLIP Img Score(↑) | CSD(↑) | PrefDisc(↑) |
> |----------------|---------|---------|-------------------|--------|-------------|
> | IP-Adapter     | 206.96  | 0.27    | 0.68              | 44.81  | 68.94       |
> | Instant Style  | 189.40  | 0.26    | 0.76              | 54.46  | 69.40       |
> | Style Align    | 200.31  | 0.25    | 0.68              | 41.30  | 68.39       |
> | Bagel          | 190.11  | 0.24    | 0.71              | 49.35  | 56.22       |
> | ViPer          | 225.63  | 0.50    | 0.70              | 46.10  | 53.61       |
> | EasyRef        | 199.34  | 0.25    | 0.74              | 51.09  | 65.80       |
> | PrefGen (Ours) | 189.34  | 0.25    | 0.77              | 57.13  | 76.78       |
>
> On this real-human subset, PREFGEN continues to outperform all baselines, achieving the highest CLIP-Img, CSD, and PrefDisc scores while maintaining competitive FID and CMMD. Notably, CMMD measures distribution-level alignment between generated samples and real user-preferred image, providing a complementary signal beyond pixel-level realism and reflecting how well the model matches preference-conditioned data distributions. Moreover, CSD (Style Distance) measures how well the generated images preserve aesthetic attributes such as color tone, texture, and artistic rendering observed in user-liked images, while PrefDisc provides an additional proxy for preference alignment by estimating whether a generated image aligns with patterns typically favored in human-annotated datasets.
> The consistently stronger CSD and PrefDisc results, together with improvements in CLIP-Img, indicate that PREFGEN more faithfully preserves both stylistic and semantic aspects of real human preferences, rather than overfitting to synthetic agent patterns.
>
> **Weakness 2: Implementation Complexity and Reproducibility**
>
> We understand the reviewer’s concern regarding implementation complexity. While our framework contains multiple conceptual components, the multimodal probing step (Step 2) is performed only once to analyze the MLLM backbone and is not part of the routine training loop.
> After this one-time analysis, the actual training pipeline is lightweight and involves only two objectives: (i) a distribution alignment loss applied exclusively to a lightweight adapter module, and (ii) the standard diffusion denoising loss.
> In practice, the overall implementation and training complexity are comparable to existing MLLM + adapter + diffusion pipelines. To support reproducibility, we will release the full codebase, dataset generation scripts, prompt templates, and all hyperparameter settings after the review period.

---

> ### Author Response · Authors · 2025-11-26
>
> **Weakness 3: Interpretability of the Hierarchical Latent Decomposition**
>
> We thank the reviewer for raising this important concern. To move beyond heuristic interpretability, we conduct **controlled swapping and ablation experiments under fixed prompts** to test whether the hierarchical decomposition captures functionally distinct factors, rather than arbitrary engineering artifacts.
>
>
> (1) In the controlled **swapping experiments**, we replace e_sem, e_core, and e_img across users while keeping the prompt fixed. Some examples are shown in [Figure 8](https://anonymous.4open.science/r/003547/fig8.png).
>
> |                | CLIP Text (↑)| CSD (↑) |
> |----------------|--------------|---------|
> | PrefGen (Ours) | 25.83        | 59.22   |
> | Swap e_sem     | 25.56        | 58.29   |
> | Swap e_img     | 25.76        | 53.27   |
> | Swap e_core    | 25.63        | 58.99   |
>
>
>
> Swapping e_sem causes the largest degradation in CLIP-Text score (25.83 → 25.56), indicating its primary role in **semantic alignment**, while swapping e_img leads to the most severe drop in CSD (59.22 → 53.27), confirming its role in **fine-grained stylistic control**. In contrast, swapping e_core produces **consistent but distributed degradation across both metrics**, suggesting that it encodes **stable, user-level preference traits that are orthogonal to purely semantic or purely stylistic factors**.
>
>
> Importantly, this interpretation is not post-hoc. In a **multi-user  linear probing experiment**, e_core achieves the **highest accuracy for user identity classification**, while e_sem is most predictive for preference discrimination, providing task-level evidence that these embeddings encode **distinct and complementary information**. We further quantify this difference through clustering metrics (ARI, NMI, AMI, V-measure), where e_core consistently exhibits stronger user-wise structure than e_sem (Appendix D.6).
>
>
>
>
> | Metric            | e_sem  | e_core |
> |-------------------|--------|--------|
> | ARI (↑)           | 0.3641 | 0.497  |
> | NMI (↑)           | 0.6497 | 0.7024 |
> | AMI (↑)           | 0.5332 | 0.6017 |
> | Homogeneity (↑)   | 0.6538 | 0.7082 |
> | Completeness (↑)  | 0.6456 | 0.6968 |
> | V-measure (↑)     | 0.6497 | 0.7024 |
>
>
>
>
>
> (2) Ablation Study:
>
> | e_img | e_sem | e_core | Alignment loss   | FID(↓) | CMMD(↓) | CLIP Img (↑) | CSD (↑) | PrefDisc (↑) |
> |-------|-------|--------|------------------|--------|---------|--------------|---------|--------------|
> | ✓     |       |        | —                | 151.65 | 0.64    | 72.76        | 45.03   | 59.77        |
> | ✓     | ✓     | ✓      | —                | 152.84 | 0.32    | 75.30        | 54.44   | 74.68        |
> |       | ✓     |        | L_MMD            | 163.35  | 0.41    | 64.25       | 31.16  | 55.98       |
> |       | ✓     | ✓      | L_MMD            | 159.72  | 0.41    | 69.21       | 37.47  | 56.83       |
> | ✓     | ✓     | ✓      | L_MMD (ours)     | 143.79 | 0.25    | 76.03        | 59.22   | 81.86        |
>
>
> Consistent with these findings, our ablation study shows that removing any component degrades both semantic fidelity and style coherence, and that only the **full decomposition** (e_img + e_core + e_sem) combined with MMD-based distribution alignment achieves simultaneous improvements across all metrics.

---

### Author Response · Authors · 2025-11-26

We sincerely thank all reviewers for their thoughtful and constructive feedback, and are encouraged by their consistent recognition of our key contributions. Reviewers highlighted that **PrefGen introduces an elegant multimodal framework grounded in a systematic analysis of MLLM-derived embeddings**, offering strong conceptual clarity and technical novelty (ovRe, P4d3). In particular, the **MMD-based distribution alignment loss** was recognized as a well-motivated and carefully justified design choice (ovRe, YgBn, P4d3).

Reviewers also emphasized the significance of our **data and evaluation contributions**, including the construction of a large-scale benchmark and human-centered evaluations (YgBn, P4d3). They further appreciated the **comprehensive experimental comparisons and in-depth analyses**, covering diverse strong baselines and systematic ablations (ovRe, YgBn, P4d3). Finally, reviewers commended the clear presentation and high-quality figures, as well as the demonstration of **practical use cases beyond generic image generation**, such as product and character design, highlighting the broader applicability of PrefGen (n3n1, P4d3).

We address all questions raised by the reviewers below and have updated the manuscript accordingly.

---

### Meta-Review · Area_Chair_JWrv · 2026-01-06

**Summary:**

The reviewers' concerns focus on 1) reliance of synthetic data, 2) complexity and reproducibility, 3) theoretical grounding and justification, and 4) evaluation gap.

1. Reviewers ovRe and YgBn express concern that the dataset relies on virtual preferences, potentially failing to capture the true human preference. Reviewers P4d3 and n3n1 have doubts on the generalizability of the proposed method.

2. Reviewers ovRe and P4d3 raise concerns on the complexity of the method. Reviewer P4d3 points out that the analysis on inference overhead is missing.

3. Reviewers ovRe and n3n1 question the theoretical basis for decomposing preferences into separate embeddings. Deeper explanations are requested. Reviewer YgBm asks for the motivation behind the use of MMD.

4. Reviewer YgBn requests more robust human evaluations in addition to the reported metrics since aesthetic is inherently subjective. Reviewer n3n1 points out that the generated results seem to skew towards style and color instead of semantic preference.

**Reviewer Concerns:**

Many concerns, including the reliance on synthetic data, theoretical justification, computational overhead, are addressed. There are outstanding concerns including pipeline complexity and justification of MMD.

**Reviewer Scores:**

This paper receives initial ratings of (4, 4, 4, 8). There are no replies from the reviewers, but given that many concerns are resolved, the AC anticipates final scores of (5, 5, 5, 8). Given the marginal ratings, the AC read the paper and review carefully, and share similar concerns of the reviewers. The authors are advised to modify the paper based on the review. A rejection is recommended.

---

### Decision · Program_Chairs · 2026-01-26

Reject